



# Impact of reactive surfaces on the abiotic reaction between nitrite and ferrous iron and associated nitrogen and oxygen isotope dynamics

Anna-Neva Visser[1,4], Scott D. Wankel[2], Pascal A. Niklaus[3], James M. Byrne[4], Andreas A. Kappler[4], Moritz F. Lehmann[1]

[1]Department of Environmental Sciences, Basel University, Bernoullistrasse 30, 4056 Basel, Switzerland

[2]Woods Hole Oceanographic Institution, Woods Hole, 360 Woods Hole Rd, MA 02543, USA

[3]Department of Evolutionary Biology and Environmental Studies, University of Zürich, Winterthurerstrasse 190, 8057 Zürich, Switzerland

[4]Department of Geosciences, Tübingen University, Hölderlinstrasse 12, 72074 Tübingen, Germany

*Correspondence to*: Anna-Neva Visser (a.visser@unibas.ch)

**Abstract.** Anaerobic nitrate-dependent Fe(II) oxidation (NDFeO) is widespread in various aquatic environments, and plays a major role in iron and nitrogen redox dynamics. However, evidence for truly enzymatic, autotrophic NDFeO remains limited, with alternative explanations involving coupling of heterotrophic denitrification with abiotic oxidation of structurally-bound or aqueous Fe(II) by reactive intermediate N species (chemodenitrification). The extent to which chemodenitrification is caused, or enhanced, by *ex vivo* surface catalytic effects has, so far, not been directly tested. To determine whether the presence of either a Fe(II)-bearing mineral or dead biomass (DB) catalyses chemodenitrification, two different sets of anoxic batch experiment were conducted: 2 mM Fe(II) was added to a low-phosphate medium, resulting in the precipitation of vivianite $(Fe_3(PO_4)_2)$, to which later 2 mM nitrite $(NO_2^-)$ was added, with or without an autoclaved cell suspension (~$1.96 \times 10^8$ cells ml$^{-1}$) of *Shewanella oneidensis* MR-1. Concentrations of nitrite, nitrous oxide $(N_2O)$ and iron $(Fe^{2+}, Fe_{tot})$ were monitored over time in both setups to assess the impact of Fe(II) minerals and/or DB as catalysts of chemodenitrification. In addition, the natural-abundance isotope ratios of $NO_2^-$ and $N_2O$ ($\delta^{15}N$ and $\delta^{18}O$) were analysed to constrain associated isotope effects. Up to 90% of the Fe(II) was oxidized in the presence of DB, while only ~65% were oxidized under mineral-only conditions, suggesting an overall lower reactivity of the mineral-only setup. Similarly, the average $NO_2^-$ reduction rate in the mineral-only experiments ($0.004 \pm 0.003$ mmol L$^{-1}$ day$^{-1}$) was much lower compared to experiments with mineral plus DB ($0.053 \pm 0.013$ mmol L$^{-1}$ day$^{-1}$), as was $N_2O$ production ($204.02 \pm 60.29$ nmol/L*day). The $N_2O$ yield per mole $NO_2^-$ reduced was higher in the mineral-only setups (4%) compared to the experiments with DB (1%), suggesting the catalysis-dependent differential formation of NO. N-$NO_2^-$ isotope ratio measurements indicated a clear difference between both experimental conditions: in contrast to the marked $^{15}N$ isotope enrichment during active $NO_2^-$ reduction ($^{-15}\varepsilon_{NO2} = +10.3‰$) observed in the presence of DB, $NO_2^-$ loss in the mineral-only experiments exhibited only a small N isotope effect ($<+1‰$). The nitrite O isotope effect was very low in both setups ($^{18}\varepsilon_{NO2} <1‰$), most likely due to substantial O isotope exchange with ambient water. Moreover, during the low-turnover conditions (i.e., in the mineral-only experiments, as well as initially in experiments with DB), the observed nitrite isotope systematics suggest, transiently, a small inverse isotope effect (i.e., decreasing nitrite $\delta^{15}N$ and $\delta^{18}O$





with decreasing concentrations), possibly related to transitory surface complexation mechanisms. Site preference (SP) of the
$^{15}$N isotopes in the linear $N_2O$ molecule for both setups ranged between 1 to 7‰, notably lower than previously reported for
chemodenitrification. Our results imply that chemodenitrification is dependent on the available reactive surfaces, and that the
$NO_2^-$ (rather than the $N_2O$) isotope signatures may be useful for distinguishing between chemodenitrification catalysed by
minerals, chemodenitrification catalysed by dead microbial biomass, and possibly true enzymatic NDFeO.

## 1. Introduction

Iron (Fe) is essential for all living beings and its biogeochemical cycling has been studied extensively (Expert, 2012; Lovley,
1997). Although Fe is ubiquitous in most environments, it is not always bioavailable (Andrews et al., 2003; Ilbert and
Bonnefoy, 2013), and microorganisms must often cope with Fe limitation in their respective environments (Braun and Hantke,
2013; Ilbert and Bonnefoy, 2013). This is especially true at circumneutral pH and oxic conditions, where Fe(II) is quickly
oxidized by $O_2$ and thus only present as poorly soluble Fe(III)(oxyhydr)oxides (Cornell and Schwertmann, 2003; Stumm and
Sulzberger, 1992). In contrast, under anoxic conditions, Fe is mainly present as either dissolved $Fe^{2+}$ or as mineral-bound Fe(II)
in iron phosphates or carbonates (Charlet et al., 1990; Luna-Zaragoza et al., 2009). Here, microbes use electron acceptors other
than $O_2$ for respiration (He et al., 2016; Lovley, 2012; Straub et al., 1996). One redox pair that has been proposed to be exploited
by microbes under anoxic conditions is $NO_3^-/Fe^{2+}$, through a mechanism known as nitrate-dependent Fe(II) oxidation (NDFeO)
(Ilbert and Bonnefoy, 2013; Straub et al., 1996). Over the past two decades, several microorganisms have been investigated
and reported to be either chemolithoautotrophic or -mixotrophic nitrate-dependent Fe(II)-oxidising bacteria (e.g. *Acidovorax*
*delafieldii* strain 2AN, *Pseudogulbenkiania ferrooxidans* strain 2002) (Chakraborty et al., 2011; Weber et al., 2009). It has
been suggested that extracellular electron transfer (EET) might play a major role in NDFeO (Liu et al., 2018). Particularly in
the presence of high levels of extracellular polymeric substances (EPS) (Klueglein et al., 2014; Zeitvogel et al., 2017), which
can act as electron shuttles, EET may indeed provide a plausible explanation for the observed Fe(II) oxidation in these cultures
(Liu et al., 2018). The existence of such an electron transfer would imply that NDFeO is not necessarily a completely
biologically catalysed reaction. Indeed, to date, genetic evidence that supports this metabolic capacity of the studied
microorganisms remains lacking (Price et al., 2018), and biogeochemical evidence is rare and putative. The latter is mostly
based on experiments with the chemolithoautotrophic culture KS, a consortium of four different strains, including a relative
of the microaerophilic *Sideroxydans/Gallionella*. This enrichment culture has been shown to be able to oxidize Fe(II) without
the addition of any organic co-substrates (Tominski et al., 2018). Tian et al. (2020) confirmed that *Gallionellaceae* are able to
perform autotrophic Fe(II)-dependent denitrification. Another more indirect line of evidence includes results from slurry
microcosm experiments with marine coastal sediments. In these experiments, Fe(II) oxidation was still detected even after all
organics of the sediments were consumed and only nitrate was left (Laufer et al., 2016). With regards to other studies where
NDFeO was initially thought to be performed by autotrophs (Chakraborty et al., 2011; Weber et al., 2009), it was subsequently
shown that the microbes rely on an organic co-substrate and must in fact be considered mixotrophic (Klueglein et al., 2014;





Muehe et al., 2009). Yet, the exact mechanism promoting NDFeO is still not fully understood. Considering that all putative
NDFeO strains were grown under high (up to 10 mM) nitrate ($NO_3^-$) and Fe(II) concentrations, and accumulated up to several
mM nitrite ($NO_2^-$) from enzymatic $NO_3^-$ reduction, it was suggested that the observed Fe(II) oxidation in these pure cultures
may be due to the abiotic side reaction between the generated $NO_2^-$ and Fe(II) (Buchwald et al., 2016; Prakash Dhakal, 2013;
Klueglein et al., 2014). This abiotic reaction between $NO_2^-$ and Fe(II) is known as chemodenitrification (Equation 1) and is
proposed to lead to an enhanced production of $N_2O$ (Anderson and Levine, 1986; Buchwald et al., 2016; Jones et al., 2015).

$$4Fe^{2+} + 2NO_2^- + 5H_2O \rightarrow 4FeOOH + N_2O + 6H^+ \qquad \Delta G° = -128.5 \; ^{kJ}/_{mol} \qquad (1)$$

Several studies have noted that the presence of reactive surfaces may enhance the abiotic reaction (Heil et al., 2016; Sorensen
and Thorling, 1991). For example, Klueglein and Kappler (2013) tested the impact of goethite on Fe-coupled
chemodenitrification in the presence of high Fe(II) and $NO_2^-$ concentrations, and confirmed the concentration dependency of
this reaction with regard to both species (Van Cleemput and Samater, 1995). Possible catalytic effects (e.g. by reactive surfaces
and/or organic matter) were not tested specifically in these studies. Yet, multiple factors have been shown to affect the abiotic
reaction between $NO_2^-$ and Fe(II) and may need to be considered (i.e.. pH, temperature, $Fe^{2+}$ concentrations, solubility of
Fe(III)(oxyhydr)oxides, crystallinity of Fe(II) minerals, other metal ion concentrations and catalytic effects) (Van Cleemput
& Samater, 1995; Klueglein & Kappler, 2013; Ottley et al., 1997). In addition, the presence of organic compounds can lead to
the abiotic reduction of $NO_2^-$ to NO (Van Cleemput and Samater, 1995; McKnight et al., 1997; Pereira et al., 2013).
Given the complex controls and potential interaction between Fe(II) and various nitrogenous compounds, including
intermediates, it may be an oversimplification to state that Fe(II) oxidation observed in previous laboratory setups is solely
caused by the abiotic reaction with $NO_2^-$, and not, for example, stimulated by reactive surfaces (minerals, organic-detritus) or
by nitric oxide (NO), a highly reactive intermediate not easily quantified in anoxic experiments. In order to better understand
the factors that may control chemodenitrification of $NO_2^-$, this study focuses on the possible catalytic surface effects induced
by a Fe(II) mineral phase or DB. Furthermore, microbial cells, dead biomass, or detrital waste products might not only provide
additional reactive surface area, but may directly react with $NO_2^-$ to form NO.
Stable isotopes of both N and O ($\delta^{15}N$ and $\delta^{18}O$) offer a promising approach to further elucidate the mechanism of NDFeO,
and also to more generally expand our understanding of chemodenitrification. The N and O isotopic composition of
nitrogenous compounds (e.g., $NO_3^-$, $NO_2^-$, and $N_2O$) has been used to gain deeper insights into various N turnover processes
(Granger et al., 2008; Jones et al., 2015). The dual $NO_2^-$ (or $NO_3^-$) isotope approach is based on the fact that specific N-
transformation processes – biotic or abiotic – are associated with specific N and O isotope fractionation (i.e., isotope effect).
In general, enzymatic processes promote the more rapid reaction of lighter N and O isotopologues, leaving the remaining
substrate pool enriched in the heavier isotopes (i.e., $^{15}N$, $^{18}O$) (Granger et al., 2008; Kendall & Aravena, 2000; Martin &
Casciotti, 2017). Only a few studies exist that have looked into the isotope effects of chemodenitrification and reports on the
associated isotope effects are variable. Consistent with what we know from biological denitrification, chemodenitrification
experiments with 10 mM Fe(II) and $NO_2^-$, with very high reaction rates, revealed a significant increase in the $\delta^{15}N$ (up to 40‰)

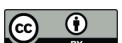



and $\delta^{18}O$ (up to 30‰) $NO_2^-$ values, corresponding to an overall N and O isotope effect of $^{15}\varepsilon$ 18.1 ± 1.7‰ and $^{18}\varepsilon$ 9.8 ± 1.8‰,
as well as a $\Delta^{15}N$ (i.e., the difference between $\delta^{15}NO_2^-$ and $\delta^{15}N_2O$) of 27 ± 4.5‰ (Jones et al., 2015). This suggests that
coupled N and O isotope measurements hold the potential to disentangle abiotic and biotic $NO_2^-$ reduction in the presence of
Fe(II). Here, in order to expand the limited dataset on the isotope effects of abiotic Fe(II)-coupled denitrification, and in turn
to lay the groundwork for using $NO_3^-/NO_2^-$ N and O isotope measurements to unravel the mechanism behind NDFeO, we
studied the N and O isotope dynamics of $NO_2^-$ reduction and $N_2O$ production during abiotic reaction of $NO_2^-$ with Fe(II). As
the extent of the formation of various Fe(III)(oxyhydr)oxides has been previously reported to enhance chemodenitrification
dynamics (Chen et al., 2018; Sorensen and Thorling, 1991), we also followed mineral alteration during chemodenitrification
in order to identify possible reaction patterns. A specific goal in this context was to assess the impact of Fe(II) precipitates
and/or dead biomass as catalytic agents during Fe(II)-associated chemodenitrification, as well as potential mineral
transformation processes associated with the abiotic oxidation of Fe(II) via reactive $NO_x$ species.

## 2.   Material and Methods

### 2.1.   General experimental setup

For all experiments, anoxic low phosphate medium (1.03 mM $KH_2PO_4$, 3.42 mM NaCl, 5.61 mM $NH_4Cl$, 2.03 mM $MgSO_4·7$
$H_2O$ and 0.68 mM $CaCl_2·2 H_2O$, with a 7-vitamin (Widdel & Pfennig, 1981) and a SL-10 trace element solution (Widdel et
al., 1983); 22 mM bicarbonate buffered) was prepared. The medium was dispensed with a Widdel flask in 1-l Schott bottles
and the pH for each bottle was adjusted separately by the addition of anoxic, sterile 1 M HCl. For the both setups, five different
pH values were targeted: 5.8, 6.2, 6.5, 6.9 and 7.1. After pH adjustment, Fe(II)Cl$_2$ was added to reach a concentration of ~2
mM Fe(II), and, if necessary, the pH was re-adjusted. The medium was kept for 48 h at 4°C, resulting in amorphous, green-
greyish Fe(II) precipitates. In addition, ~2 mM $NaNO_2$ and ~1 mM Na-acetate were added to the main medium stocks shortly
before 10 ml aliquots of the medium were distributed into 20 ml headspace vials (heat-sterilized) in an anoxic glove box
(MBraun, $N_2$, 100%). Acetate was added to mimic experiments, in which bacteria are cultivated (yet, acetate concentrations
did not change during incubations, underscoring that the organic acid was not involved in the observed reactions; data not
shown). All headspace vials were closed with black butyl stoppers and crimp-sealed [headspace $N_2/CO_2$ (90/10, v/v)]. All vials
were then incubated at 28°C in the dark.
*Incubations with dead-biomass – Shewanella oneidensis* MR-1, a facultatively aerobic Gram-negative bacterium, is seen as
model organism for bioremediation studies due to its various respiratory abilities (Heidelberg et al., 2002; Lies et al., 2005). It
is known to perform dissimilatory metal reduction by utilizing alternative terminal electron acceptors such as elemental sulfur,
Mn(IV), Fe(III) or $NO_3^-$. Since *S. oneidensis* produces large amounts of EPS (Dai et al., 2016; Heidelberg et al., 2002), but is
not capable of oxidizing Fe(II) (Lies et al., 2005; Piepenbrock et al., 2011) (i.e. no interference with abiotic reactions involving
Fe/chemodenitrification), we chose concentrated and sterilized *S. oneidensis* for our dead-biomass experiments. In preparation
of these experiments, *S. oneidensis* MR-1 was grown oxically on a LB (lysogeny broth) medium (10 g tryptone, 5 g yeast



extract, 10 g NaCl in 1 l DI water) in six 250 ml Erlenmeyer flasks. After 12 hrs, cultures were transferred into 50 ml Falcon
tubes and centrifuged for 25 min at 4000 rpm (Eppendorf, 5430 R). Cell-containing pellets were washed twice with oxalic acid
and centrifuged again, followed by three more washing steps with TRIS buffer prior to final resuspension in 5 ml TRIS buffer.
Pellet suspensions were pooled in a 100 ml serum bottle and autoclaved twice to ensure that all cells were killed. Before
distribution of the medium into 20 ml vials (see above), cell suspension was added to yield a cell density of ~$1.96 \times 10^8$ cell ml$^{-1}$
$^1$. Care was taken to ensure the homogenous distribution of mineral precipitates and the dead biomass.

## 2.2. Sampling and sample preparation

Incubations were run for approximately 30 days, and sampling was performed in an anoxic glove box (MBraun, N$_2$, 100%) at
five time points. For each time point, and for each pH treatment, 9 replicates were prepared. Therefore, variations between the
replicates and the different sampling time points are possible. For sampling, the headspace was quantitatively transferred into
12 ml He-purged Exetainer vials (LABCO) for N$_2$O concentration measurements. Then, 2 ml of the liquid sample were
transferred into 2 ml Eppendorf tubes, centrifuged 5 min (13400 rpm; Eppendorf, MiniSpin), followed by a 1:10 dilution of
the supernatant in 1 ml anoxic MilliQ water for NO$_2^-$ quantification. A second 100 µl aliquot was diluted 1:10 in 40 mM
sulfamic acid (SFA) for iron determination by ferrozine analysis (Granger and Sigman, 2009; Klueglein and Kappler, 2013).
The remaining supernatant was used for HPLC and NO$_2^-$ isotope analysis. Finally, the spun-down pellet was resuspended in 1
M HCl for ferrozine analysis (Stookey, 1970). All samples were stored at 4°C in the dark until further processing. The
remaining liquid samples were used for $^{57}$Fe Mössbauer spectroscopy.

## 2.3. Analytical techniques

*NO$_2^-$ concentrations* – NO$_2^-$ concentrations were quantified using standard segmented continuous-flow analytical (CFA, SEAL
Analytics) photometric techniques (Snyder and Adler, 1976). NO$_2^-$ reduction rates were calculated based on the observed net
concentration decrease ($\overline{[C]}_{t0} - \overline{[C]}_{tend}$ ±standard error) with time.
*Fe concentrations* – Fe(II) concentration was analysed using the ferrozine assay (Stookey, 1970), which was adapted for NO$_2^-$
-containing samples by Klueglein et al. (2013). Total Fe(II) concentrations were calculated as the sum of the $Fe^{2+}_{aq}$ +
$Fe(II)_{pellet}$ concentrations.
*N$_2$O concentrations* – Prior to the quantification of the N$_2$O, the sample gas was diluted (1:5) with 5.0 He. The samples were
then analysed using a gas chromatograph with an electron capture detector (GC-ECD; Agilent 7890 with micro-ECD and FID;
Porapak Q 80/100 column). GC-ECD measurements were calibrated using four standard gases containing different
concentrations of N$_2$O (Niklaus et al., 2016). N$_2$O production rates were calculated based on the observed net N$_2$O
concentration increase ($\overline{[C]}_{tend} - \overline{[C]}_{t0}$ ±standard error) with time.
*$^{57}$Fe Mössbauer spectroscopy* - For Mössbauer spectroscopic analyses, the remaining liquid samples (ca. 8 ml) were processed
inside an anoxic glove box. The entire liquid including the precipitates was passed through a 0.45 µm filter. The wet filter was





then sealed between two layers of Kapton tape and kept inside sealed Schott bottles in a freezer (-20°C) under anoxic conditions
until analysis. From the treatments with DB, samples were collected at day 0 at pH 6.8 and at the end of the experiment (~30
days) for pH 6.8 and 5.8. For the mineral-only experiment, only one sample (time point zero, pH 6.8) was analysed, as a basis
for comparison with the DB experiments (i.e., to verify whether DB has an immediate effect on the mineral phase). Taking
care to minimize exposure to air, samples were transferred from the air-tight Schott bottles and loaded inside a closed-cycle
exchange gas cryostat (Janis cryogenics). Measurements were performed at 77 K with a constant acceleration drive system
(WissEL) in transmission mode with a $^{57}$Co/Rh source and calibrated against a 7µm thick α-$^{57}$Fe foil measured at room
temperature. All spectra were analysed using Recoil (University of Ottawa) by applying a Voight Based Fitting (VBF) routine
(Lagarec and Rancourt, 1997; Rancourt and Ping, 1991). The half-width at half maximum (HWHM) was fixed to a value of
0.130 mm/s during fitting.
*Nitrite N and O isotope measurements* – The nitrogen (N) and oxygen (O) isotope composition of $NO_2^-$ was determined using
the azide method (McIlvin and Altabet, 2005). This method is based on the chemical conversion of $NO_2^-$ to gaseous $N_2O$ at a
low pH (4 to 4.5) (McIlvin and Altabet, 2005), and the subsequent analysis of the concentrated and purified $N_2O$ by gas
chromatography— isotope ratio mass spectrometry (GC-IRMS). Addition of 0.6 M NaCl to the acetic acid-azide solution was
conducted in order to minimize oxygen isotope exchange (McIlvin and Altabet, 2005). The acetic acid-azide solution was
prepared freshly every day (McIlvin and Altabet, 2005) and kept in a crimp sealed (grey butyl stopper) 50 ml serum bottle.
Sample volume equivalent to 40 nmol $NO_2^-$ was added to pre-combusted headspace vials, filled up to 3 ml with anoxic MilliQ
water, and crimp-sealed. Then, 100 µl of the acetic acid/azide solution was added. After ~7 hrs, 100 µl of 6 M NaOH was
added to stop the reaction. Until isotope analysis by a modified purge and trap gas bench coupled to CF-IRMS (McIlvin and
Casciotti, 2010), the samples were stored upside down in the dark. Nitrite isotope standards (N-7373 and N-10219; Casciotti
& McIlvin, 2007) were prepared on the day of isotope analysis and processed the same way as samples. N and O isotope data
are expressed in the common δ notation and reported as per mille deviation (‰) relative to AIR $N_2$ and VSMOW, respectively
$((\delta^{15}N = ([^{15}N]/[^{14}N])_{sample} / [^{15}N]/[^{14}N]_{air\_N2} - 1) \times 1000‰$ and $\delta^{18}O = ([^{18}O]/[^{18}O]_{sample} / [^{18}O]/[^{16}O]_{VSMOW} - 1) \times 1000‰)$.
*$N_2O$ N and O isotope measurements* – Triplicate 20-nmol samples of $N_2O$ were injected into 20 ml headspace vials that were
flushed before for 5 hrs with 5.0 He (injection volumes according to the $N_2O$ concentrations determined before). The $N_2O$ was
then analysed directly using CF-IRMS (see above). Two standard gases with known $\delta^{15}N$ and $\delta^{18}O$ values were analysed along
with the samples, namely FI.CA06261 ($\delta^{15}N$: -35.74‰, $\delta^{15}N^\alpha$: -22.21‰, $\delta^{15}N^\beta$=-49.28‰, $\delta^{18}O$: 26.94‰) and FI.53504 ($\delta^{15}N$:
48.09‰, $\delta^{15}N^\alpha$: 1.71‰, $\delta^{15}N^\beta$=94.44‰, $\delta^{18}O$: 36.01‰) (provided by J. Mohn, EMPA; e.g. Mohn et al., 2014). The gases
were calibrated on the Tokyo Institute of Technology scale for bulk and site-specific isotopic composition (Ostrom et al., 2018;
Sakae Toyoda et al., 1999). Ratios of m/z 45/44, 46/44 and the 31/30 signals were used to calculate values of $\delta^{15}N^{bulk}$
(referenced against AIR-$N_2$), $\delta^{18}O$ (referenced against V-SMOW), and site-specific $\delta^{15}N^\alpha$, $\delta^{15}N^\beta$ based on Frame and Casciotti
(2010) . Site preference (SP) was calculated as $\delta^{15}N^\alpha – \delta^{15}N^\beta$ (Sutka et al., 2006; Toyoda and Yoshida, 1999).





## 2.4. Pourbaix diagram


In order to predict the stability and behaviour of the N- and Fe(II)-bearing chemical species in the same system, a Pourbaix
(Eh-pH) diagram was constructed (Delahay et al., 1950) as a valuable tool to predict possible reactions and speciation of end
products under different experimental conditions. To calculate the enthalpies for the stepwise reduction of nitrite during
denitrification, as well as Fe(II) oxidation reactions, standard enthalpy values were taken from different references (Table S1).
The Pourbaix diagram presented in the discussion was devised using concentrations measured during the experiments
performed for this study.

## 3. Results


## 3.1. Chemodenitrification kinetics


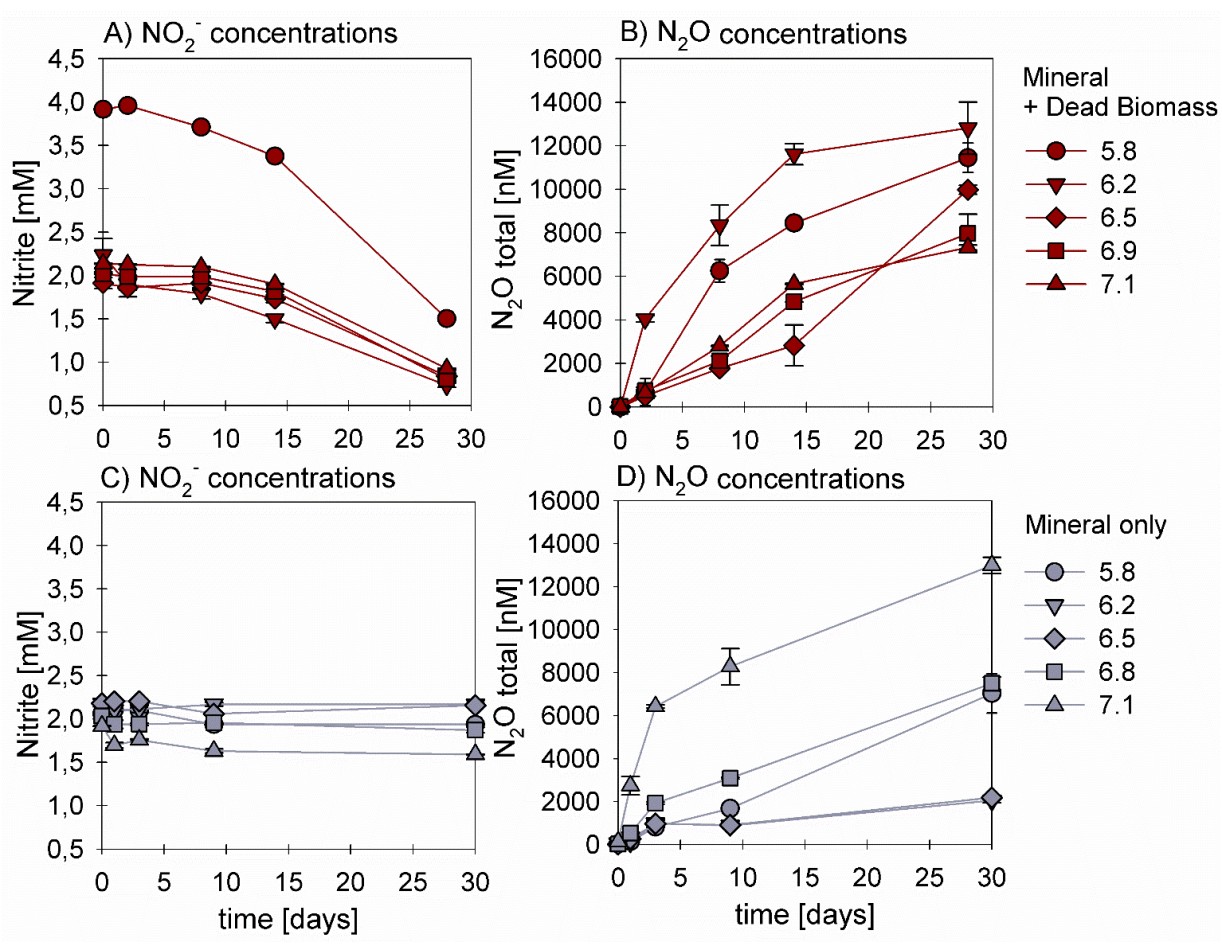

**Figure 1: Nitrite reduction (A, C) and N₂O production (B, D) over time in the mineral + dead biomass (red) and mineral-only (grey)**
**setups over time and at different pH. Please note that at pH 5 twice the amount of nitrite was accidently introduced. Standard error**
**calculated from biological replicates (n = 9) is represented by the error bars.**
In the presence of DB, $NO_2^-$ reduction rates were much higher compared to the mineral-only setup (Figure 1 A, C), with up to
~60% of the initially amended $NO_2^-$ being transformed during the incubation period, independent of the pH. The addition of
DB led to a decrease in $NO_2^-$ concentrations from 2 mM to ~0.7 mM (Figure 1 A). The pH 5.8 treatment (unintentionally
amended with 2x $NO_2^-$) also showed a similar fractional reduction. In the mineral-only setups the decrease in $NO_2^-$
concentration was rather moderate and ranged between 0.3 (pH 7) and 0.1 mM (at lower pH) (Figure 1 C). In all treatments,
$N_2O$ was produced but accounted for a maximum of only 0.7% of the $NO_2^-$ consumed. The final $N_2O$ yield per mole $NO_2^-$
reduced tended to be lower in the mineral plus DB versus the mineral-only amended setups for most of the pH (Figure 1 B vs.
D). Highest $N_2O$ production was observed at circumneutral pH (7.1) in the mineral-only setup, while maximum final $N_2O$
concentrations were observed at lower pH (6.2) in the incubations with DB (Figure 1 B). A systematic pH effect, however,
could not be discerned. $Fe(II)_{total}$ concentrations rapidly decreased in both setups. In the presence of DB, $Fe(II)_{total}$ oxidation
was almost complete (Figure 2A), independent of the pH, whereas in the mineral-only experiment, $Fe(II)_{total}$ decreased during
the first 5-10 days but then seemed to reach a steady state (Figure 2 B). At pH 6.8 and 5.8, only 40% of the $Fe(II)_{total}$ was
oxidized, whereas at the other pH up to 80% of the $Fe(II)_{total}$ initially amended was oxidized. Total Fe decreased over time
(Figure S2).

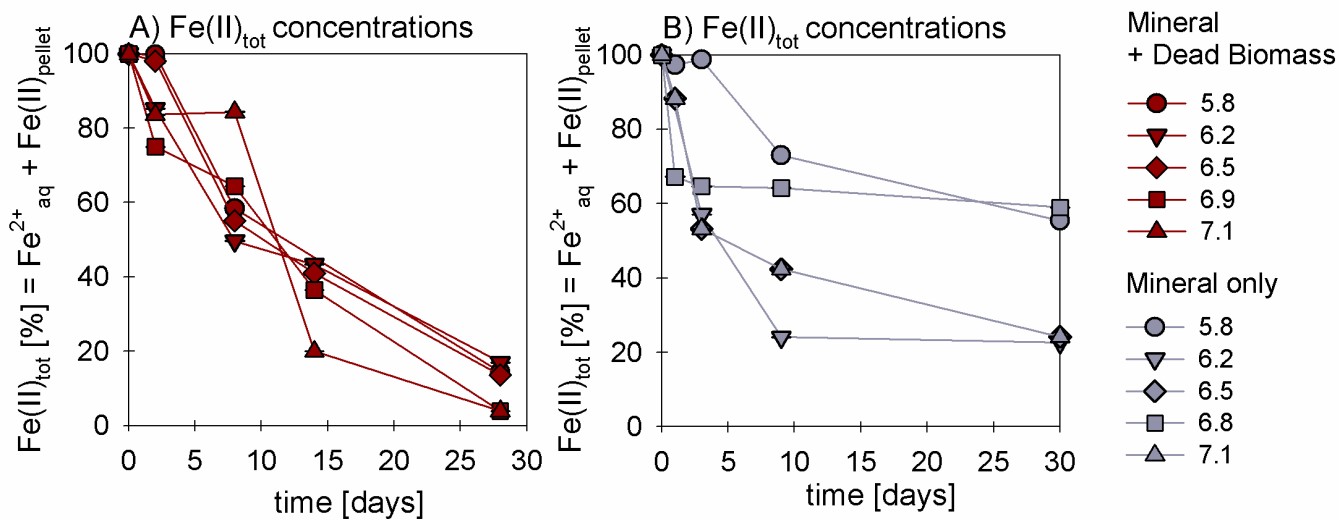


**Figure 2: Oxidation of total Fe(II) over time given (reported as % of initial concentration) in the mineral + dead biomass amended**
**(red) and the mineral-only setup (grey), tested at different pH. Standard error calculated from biological replicates (n = 9) is**
**represented by the error bars.**

Average rates for $NO_2^-$ reduction and $N_2O$ production at pH 6.8 were calculated (Table 1). Rates were calculated per day and
again these results emphasize that the amendment of dead biomass increased the rates by ~92%. Although not complete, Fe(II)
oxidation in the presence of DB was also more pronounced leading to only 10.5±2.8% Fe(II) remaining compared to the
mineral-only setup in which 37.1±8.2% Fe(II) remained. To complement the colorimetric data, $^{57}$Fe Mössbauer spectroscopy
was performed and data are presented in detail in the next section.



**Table 1: Chemodenitrification kinetics and mineral transformation during mineral + dead biomass as well as the mineral only**
**experiments. $T_{ini}$ values represent means calculated by summarizing results across all pH ± standard error. Overall**
**reduction/production rates are calculated by subtracting $\overline{[C]}_{t0} - \overline{[C]}_{tend}$ ±standard error/ $\overline{[C]}_{tend} - \overline{[C]}_{t0}$ ±standard error,**
**respectively and are given per day. Fe(III) values are calculated by using [57]Fe Mössbauer spectroscopy data. Mineral phases were**
**also identified by using [57]Fe Mössbauer spectroscopy with spectra collected at 77 K. Mineral-only sample taken after 28 days was**
**inadvertently destroyed prior to Mössbauer measurement.**

|  | **Mineral + Dead Biomass** | **Mineral-only** |
|---|---|---|
| **$NO_2^-$ reduction ($\bar{X}$)** | 0.053 ±0.013 mmol L$^{-1}$ day$^{-1}$ | 0.004 ±0.003 mmol L$^{-1}$ day$^{-1}$ |
| **$N_2O$ production ($\bar{X}$)** | 353.50 ±32.91 nmol L$^{-1}$ day$^{-1}$ | 204.02 ±60.29 nmol L$^{-1}$ day$^{-1}$ |
| **Fe(II)$_{total}$ remaining ($\bar{X}$)** | 10.54 ±2.77% | 37.08 ±8.23% |
| **Fe(III) after $NO_2^-$ addition** | 7.4% | 9.9% |
| **Fe(III) after 28 days** | 48.7% | * |
| **Mineral phase $t_{ini}$** | Vivianite | Vivianite |
| **Mineral phase $t_{end}$** | Vivianite/Ferrihydrite | * |

*\* Mössbauer sample lost*

**3.2. Fe mineral analysis**
[57]Fe Mössbauer spectroscopy was used to quantify structural Fe(II) and Fe(III) contents of the samples and identify differences
in mineralogy under the different reaction conditions. The hyperfine parameters of the mineral phases in in the mineral-only
setup at $t_{initial}$ (pH 6.84) are dominated by Fe(II) doublets (Figure 3 A, QSD Sites 1 and 2), which most closely match that of a
vivianite spectrum (Muehe et al., 2013; Veeramani et al., 2011). There is a small component with low centre shift and
quadrupole splitting, indicative of Fe(III), which accounts for ~10% of the spectral area (Figure 3 A, QSD Site 3). This suggests
some minor oxidation occurred, potentially during transfer of sample into the spectrometer. The mineral phases in the DB-
amended setup at $t_{initial}$ (pH 6.89) shows very close approximation to the abiotic mineral-only setup, though with slightly less
Fe(III) (~7.5% of the spectral area) (Figure 3 B, QSD Site 2). Precipitates analysed at the end of the DB-amended experiment
(Day 28) show that at pH 6.89, the vivianite phase still dominates (Figure 3 C, QSD Sites 1 and 2), however, the Fe(III)
component is now much more prominent (Figure 3 C, QSD Site 3), and suggests the formation of a poorly crystalline/short-
ranged ordered mineral such as ferrihydrite (Cornell and Schwertmann, 2003). At the lowest pH (5.78) and in the presence of
DB, the pattern of the precipitates is completely dominated by one doublet (Figure 3 C, QSD Site 1), with hyperfine parameters
corresponding to a poorly ordered Fe(III) mineral such as ferrihydrite (Cornell and Schwertmann, 2003). Unfortunately, the
mineral-only sample taken after 28 days was lost and can therefore not be used for further elucidations. Detailed fitting results
of the [57]Fe Mössbauer spectroscopy are provided in Table 2.






**Figure 3:** $^{57}$Fe Mössbauer spectra collected at 77 K for (A) the mineral only setup precipitates at day 0 and pH 6.84, (B) the mineral + dead biomass amended setup precipitates at day 0 at pH 6.89, (C) the mineral + dead biomass amended setup precipitates at day 28 and (D) the mineral + dead biomass amended setup precipitates at day 28 at pH 5.78. Full lines represent the calculated spectra and their sums. Colours of the fits represent the corresponding Fe phase and thus vary between the graphs: Fe(II) doublets (A, C – QSD Sites 1 and 2, B – QSD Sites 1 and 3) closely match the spectra known for vivianite. Minor amounts of Fe(III) are present at day 0 in both, the mineral-only and DB-amended setups (A/B QSD Site 3/2). Single doublets shown in C (QSD Site 3) and D (QSD Site 1) correspond to a poorly ordered Fe(III) mineral such as ferrihydrite.







**Table 2: Fitting results of Mössbauer spectroscopy. CS – centre shift, QS – quadrupole splitting, R.A. – Relative abundance**
**determined by integration under the curve, Chi$^2$ – goodness of fit; sample collection took place at $t_{ini}$ – initial timepoint and $t_{end}$ –**
**end timepoint; MO = mineral-only, MDB = mineral + dead biomass.**

| Sample | Temp | Phase | CS | QS | R.A. | Error | Chi$^2$ |
|---|---|---|---|---|---|---|---|
| | [K] | | [mm/s] | [mm/s] | [%] | | |
| MO_pH6.8_$t_{ini}$ | 77 | Fe(II) | 1.32 | 2.71 | 66.0 | 23.0 | 0.55 |
| | | Fe(II) | 1.33 | 3.15 | 24.0 | 23.0 | |
| | | Fe(III) | 0.47 | 0.63 | 9.9 | 4.8 | |
| MDB_pH6.8_$t_{ini}$ | 77 | Fe(II) | 1.30 | 2.70 | 65.0 | 14.0 | 0.68 |
| | | Fe(III) | 0.49 | 0.49 | 7.4 | 3.6 | |
| | | Fe(II) | 1.36 | 3.18 | 28.0 | 15.0 | |
| MDB_pH6.8_$t_{end}$ | 77 | Fe(II) | 1.33 | 3.21 | 34.3 | 2.4 | 0.73 |
| | | Fe(II) | 1.37 | 2.44 | 17.0 | 2.8 | |
| | | Fe(III) | 0.44 | 0.89 | 48.7 | 2.4 | |
| MDB_pH5.8 _$t_{end}$ | 77 | Fe(III) | 0.49 | 0.79 | 100.0 | | 0.66 |


### 270  3.3. Nitrite and N$_2$O isotope dynamics

In the experiments with DB, the $\delta^{15}$N-NO$_2^-$ and $\delta^{18}$O-NO$_2^-$ values showed a very consistent initial ~3-4‰-decrease (from -
26‰ to -30‰ for $\delta^{15}$N and from ~+3‰ to 0‰ for $\delta^{18}$O) (Figure 4 A, B). After 5 days, the $\delta^{15}$N values started to increase again
with decreasing NO$_2^-$ concentrations, reaching final values of ~ -20‰ (Figure 4 A), whereas the concomitant increase in the
$\delta^{18}$O-NO$_2^-$ was much smaller (<1‰, Figure 4 B). The same pattern was observed for all pH levels. In mineral-only experiments,
isotope trends were quite different. In combination with far less consumption of NO$_2^-$, the $\delta^{15}$N-NO$_2^-$ values decreased
throughout the entire abiotic experiment (Figure 4 C). In contrast, the $\delta^{18}$O-NO$_2^-$ first dropped by 2‰, reaching a clear
minimum of ~0.5 to -0.5 ‰, before rapidly increasing again. Over the remaining 25 days, the $\delta^{18}$O-NO$_2^-$slowly decreased
reaching final values of ~1‰ (Figure 4 D) – similar to that of the DB treatment.



**Figure 4: δ15N (A, C) and δ18O (B, D) values for NO2- measured in the mineral + dead biomass amended (red) and the mineral-only (grey) setups over time and at different pH. Standard error calculated from biological replicates (n = 3) is represented by the error bars.**

In order to estimate the net N and O isotope fractionation for putative NO$_2^-$ reduction (in the DB-amended experiments, where we observed a clear decrease in NO$_2^-$), we plotted the NO$_2^-$ $\delta^{15}$N and $\delta^{18}$O values against the natural logarithm of the concentration of the residual NO$_2^-$ (Rayleigh plot), where the slope of the regression line approximates the N and O isotope effects, respectively (Mariotti et al., 1981). At least after the initial period, when the NO$_2^-$ $\delta^{15}$N markedly increased with decreasing NO$_2^-$ concentrations, the N isotope data are more or less consistent with Rayleigh isotope fractionation kinetics. The slope of the regression line suggests an average N isotope effect of -10.4‰ (Figure 5 A). For the mineral-only setup, no N isotope effect could be calculated, but the observed NO$_2^-$ $\delta^{15}$N trend suggest a small inverse N isotope fractionation.


Similarly, trends in $NO_2^-$ $\delta^{18}O$ of the DB experiments are not as obviously governed by normal Rayleigh fractionation
dynamics, at least not during the initial period, when the $\delta^{18}O$ decreased despite decreasing $NO_2^-$ concentrations. Considering
the $\delta^{18}O$ values only after 2 days of the incubation, the Rayleigh plot revealed an average O isotope enrichment factor of -0.5
‰ (Figure 5 B), much lower than for N. Similar to N, O-isotope Rayleigh plots for the mineral-only experiments (Figure S4)
did not exhibit coherent trends, as the fractional $NO_2^-$ depletion was minor and not consistent (mostly less than 10%). Again,
the observed $\delta^{18}O$ minimum at day 2 of the abiotic incubations suggests that processes other than normal kinetic fractionation
during $NO_2^-$ reduction were at work, which cannot be described with the Rayleigh model. If at all, the decreasing $\delta^{18}O$ values
after day 5 in the mineral-only experiments, accompanying the subtle decrease in $NO_2^-$ concentration in at least some of the
treatments, suggest a small apparent inverse O isotope effect associated with the net consumption of $NO_2^-$. Despite the different
$NO_2^-$ $\delta^{18}O$ dynamics during the course of the experiment, the final $\delta^{18}O$ of the residual nitrite was very similar in both
experimental setups, and independent of the pH.

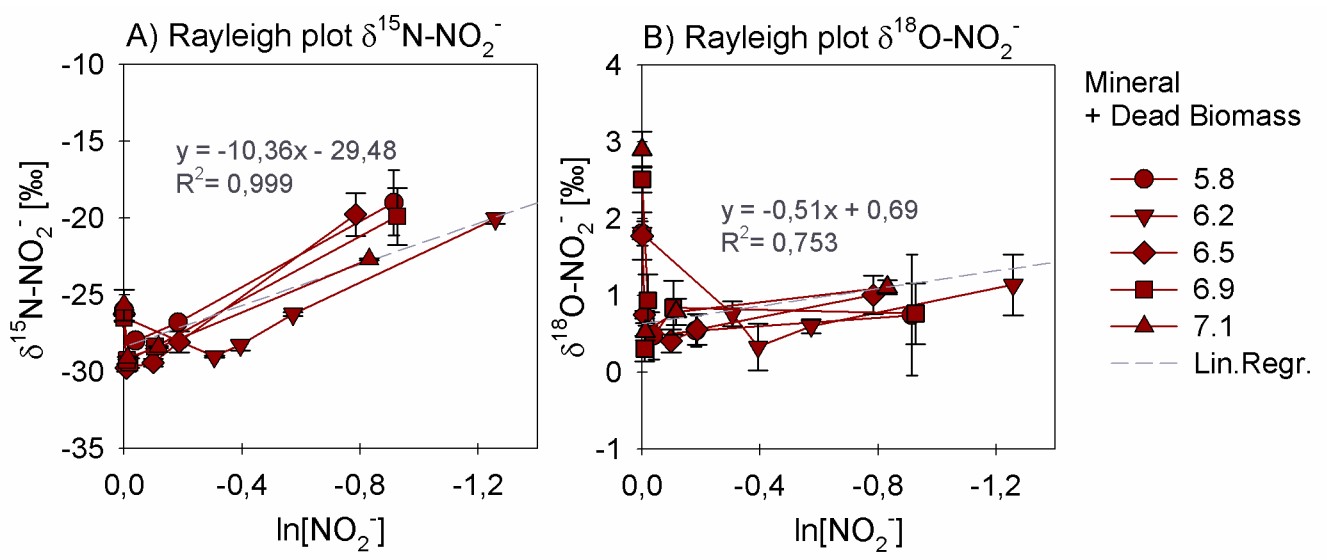


**Figure 5: Rayleigh plots for $NO_2^-$ $\delta^{15}N$ (A) and $\delta^{18}O$ (B) values measured for the mineral + dead biomass amended setups over the**
**ln of the substrate fraction remaining and at different pH. The average linear regression line was calculated starting with the lowest**
**delta values (after the initial decrease in both $\delta^{15}N$ and $\delta^{18}O$ during the initial experimental phase). Equation and $R^2$ are given in**
**grey. Standard error calculated from biological replicates (n = 3) is represented by the error bars.**

We also investigated the $N_2O$ isotope dynamics during mineral-only and DB-amended incubations. Site preference and $\delta^{15}N^{bulk}$
of the $N_2O$ produced in both experimental setups were plotted over time (Figure 5 A and B) and show, except for a few values
that require further investigation, almost no variation during the period of the experiment. Also, disregarding the rather high
and unusual (but well replicated) values already mentioned, the majority of values obtained in both setups indicate that neither
pH nor the amendment of DB seems to have had any influence on the isotopic composition of the product $N_2O$ (Figure 5 B vs.
D). Over the course of the experiment, $\delta^{15}N^{bulk}$ $N_2O$ values were around $-50\pm5‰$. SP was relatively low, ranging between 0
and a maximum of $+10‰$ (Figure 5 A, C), without any significant temporal change.




**Figure 6: Site Preference (SP; A, C) and $\delta^{15}N^{bulk}$ (B, D) values of $N_2O$ produced in experiments amended with mineral + dead biomass**
**(red) and mineral-only (grey). Standard error calculated from biological replicates (n = 3, extreme values N = 2) is represented by**
**the error bars.**


Rayleigh diagrams, in which $\delta^{15}N^{\alpha}$, $\delta^{15}N^{bulk}$ and SP of the $N_2O$ were plotted against concentrations of the reactant ($NO_2^-$)
remaining (Figure S5), confirm the similar $N_2O$ isotope dynamics in the DB vs. mineral-only setups, despite the differential
degree of $NO_2^-$ reduction (only minor in the mineral-only experiment, with f always greater 0.9) and despite the different $NO_2^-$
N and O isotope dynamics. Similarly, the dual $N_2O$ $\delta^{18}O$ vs. $\delta^{15}N^{bulk}$ signatures (with the exception of two data points; Figure
S6) were almost equivalent in both setups, implying that, although modes of $NO_2^-$ reduction clearly differ, a similar mechanism
of nitrite-reduction-associated $N_2O$ production exists in both setups. The N and O isotopic results are summarized in Table 3
(see discussion).





## 4.    Discussion and implications

### 4.1.    General evaluation of the abiotic reaction systematics

Overall, the abiotic reaction between $NO_2^-$ and Fe(II), heterogenous or homogenous, has been considered thermodynamically
favourable, and as major contributor to the global $N_2O$ budget (e.g. Jones et al., 2015; Otte et al., 2019). Previous studies on
abiotic $NO_2^-$ reduction with Fe(II) have usually been performed in the presence of rather high concentrations (>2 mM) of $NO_2^-$
and/or Fe(II), without taking into account that chemodenitrification is in fact considered to be highly concentration-dependent
(Van Cleemput and Samater, 1995). In addition, reaction dynamics were often tested under variable conditions including the
presence of different Fe(II)/Fe(III) minerals, sediments, organic materials and/or bacterial cells (Chen et al., 2018; Grabb et
al., 2017; Otte et al., 2019). Whether $NO_2^-$ indeed acts as a direct oxidant of Fe(II) at circumneutral pH or whether the reaction
requires catalysis is still a matter of debate (Kampschreur et al., 2011; Sorensen and Thorling, 1991).
Integrating concentrations that are pertinent to our experiments, we constructed a Pourbaix diagram (e.g. Delahay et al., 1950;
Minguzzi et al., 2012) (Figure 7). Based on these (simplified) thermodynamic calculations, the abiotic reaction solely driven
by the reaction of $NO_2^-$ and aqueous $Fe^{2+}$ at a pH range of 5 to 7 is not supported. Under our experimental conditions, $Fe^{2+}$ is
predicted to be oxidized by NO rather than $NO_2^-$. Considering Figure 7, an accumulation of NO at µM or even mM
concentrations would result in a downward shift of the $NO_2^-$ line. Therefore, an accumulation of NO would only lower the
reactivity between $NO_2^-$ and $Fe^{2+}$, which implies that $NO_2^-$ is not oxidizing $Fe^{2+}$. Again, this also implies that the reactivity
between $NO_2^-$ and $Fe^{2+}$ is only enhanced if NO concentrations are rather low (pM range). In order to avoid NO accumulation
and thus to enhance the abiotic reaction between $NO_2^-$ and $Fe^{2+}$, NO would need to react further (either with $Fe^{2+}$ or otherwise).
This would induce a reaction cascade, resulting in the constant reduction of $NO_2^-$ and NO, and thus in higher $N_2O$
concentrations. In contrast, if NO does accumulate as previously reported, the reaction between $NO_2^-$ and $Fe^{2+}$ would be
suppressed and only NO could be reduced further to $N_2O$, a reaction that of course also depends on gas equilibration dynamics
occurring with the headspace of the system. Nevertheless, considering all these aspects, including the fact that the $N_2O$
produced corresponds only to a minor fraction of the initial $NO_2^-$ reduced, NO acting as main oxidizing agent seems more
likely. The reaction mechanisms in this system are, however, complex and we note that this simplified thermodynamic analysis
does neglect catalytic effects that are possibly induced by reactive surfaces. The complexity of this system is further indicated
by the fact that, according to the Pourbaix diagram, a pH response towards $N_2O$ accumulation would be expected which has,
however, never been reported so far. Furthermore, testing various pH did not reveal an obvious pH effect on the reaction
dynamics. Changes in pH will most certainly affect interactions between species such as HNO, $NO_2$ and $N_2O$ and thus could
impact the reaction dynamics. In addition, the results observed in the setup biased by accidentally adding twice as much $NO_2^-$
(DB, pH 5.8) do not differ from the results of the other setups and thus might question the previously mentioned concentration
dependency (i.e. $[NO_2^-]$). It appears that, for a more detailed understanding of this redox system, the reactants/intermediates
involved and thus the specific reaction kinetics would need to be determined. Unfortunately, quantification of these
intermediates is hampered by their high reactivity, transient nature, and lack of detection techniques that can be applied in



batch culture experiments. Since low amounts (e.g., pM) of NO suffice to impact reaction dynamics and thus stimulate the
reaction between $NO_2^-$ and $Fe^{2+}$, NO quantification could be crucial to assess the environmental controls on Fe(II)-coupled
chemodenitrification. In laboratory biological denitrification experiments, accumulation of NO has been reported (Goretski
and Hollocher, 1988; Zumft, 1997) and was shown to even account for up to 40% of the initial $NO_3^-$ amended (Baumgärtner
and Conrad, 1992; Choi et al., 2006; Kampschreur et al., 2011; Ye et al., 1994; Zumft, 1997). Hence, Kampschreur et al.,
(2011) concluded that chemodenitrification is not necessarily solely caused by a single-step reaction, and proposed that the
oxidation of $Fe^{2+}$ is rather caused by a two-step mechanism. They observed an immediate formation and accumulation of NO
after $NO_2^-$ was added to $Fe^{2+}$, and as soon as a considerable fraction of the $Fe^{2+}$ was oxidized, $N_2O$ formation was detected.
Although NO and other possible intermediate (e.g. $NO_2(g)$) concentrations might not play a major role with regard to mass
balance considerations, their possible impact on the overall reaction systematics as well as the isotopic fractionation, remains
unclear.

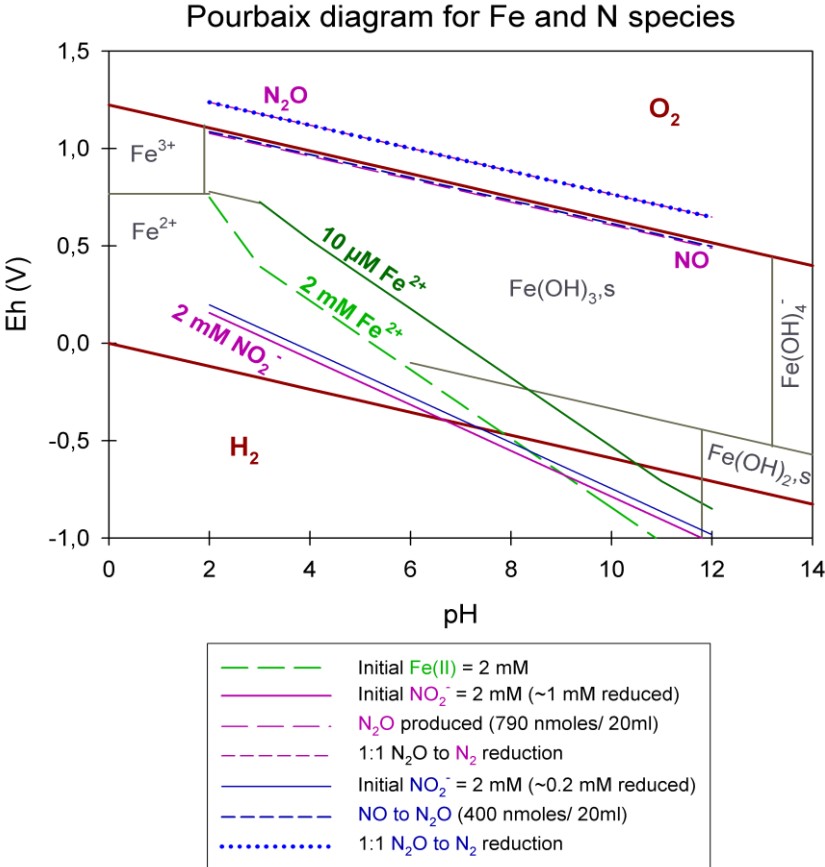

**Figure 7: Pourbaix diagram depicting an Fe and N-species based system. Overall calculations are based on the Nernst equation using**
**values taken from literature (for equation and values see table S1). Green lines represent $Fe^{2+}$ concentrations, pink lines represent**
**$NO_2^-$ reduction experiments, starting with 2 mM $NO_2^-$, resulting in the reduction of 1 mM $NO_2^-$, the production of 790 nmol /20 ml**
**$N_2O$ and a 1:1 transformation of $N_2O$ to $N_2$; blue lines represent $NO_2^-$ reduction experiments, starting with 2 mM $NO_2^-$, resulting in**
**the reduction of 0.2 mM $NO_2^-$, the production of 790 nmol /20 ml $N_2O$ and a 1:1 transformation of $N_2O$ to $N_2$. Reduction/production**
**values were taken from our results presented in 3.1.**





### 4.2. Surface catalysis of chemodenitrification

Previous studies have shown that the initial presence of either Fe(III)(oxyhydr)oxides (Coby & Picardal, 2005; Klueglein & Kappler, 2013; Sorensen & Thorling, 1991) or amorphous Fe(II) minerals (Van Cleemput and Samater, 1995) can stimulate the abiotic reaction between $NO_2^-$ and $Fe^{2+}$. As summarized in Table 1, under mineral-only conditions $NO_2^-$ reduction was significantly lower (0.004 ±0.003 mmol $L^{-1}$ $day^{-1}$) than in identical experiments containing DB, which substantially enhanced $NO_2^-$ reduction (0.053 ±0.013 mmol $L^{-1}$ $day^{-1}$). The catalytic effect of Fe minerals on the abiotic $NO_2^-$ reduction, which has been demonstrated before, seems to be amplified in the presence of DB. Relative to $NO_2^-$ reduction rates, overall final $N_2O$ yields per mole $NO_2^-$ reduced tended to be higher in the mineral-only setups. However, considering the initial $NO_2^-$ concentrations, only minor amounts of $N_2O$ were produced in both setups, raising questions about the contribution of chemodenitrification to global $N_2O$ emissions discussed by others (Grabb et al., 2017; Jones et al., 2015; Otte et al., 2019). For example, in comparison to the $N_2O$ yields in experiments where chemodenitrification was catalysed by green rust (up to 31%, Grabb et al., 2017), the amount of $N_2O$ produced in our setups is far lower (<5% of the initial $NO_2^-$).

Fe-bearing minerals are known for their high reactivity, ability to complex ligands (metals, humics) and phosphates, and surface protonation capacity via the sorption of $OH^-$ groups (Elsner et al., 2004; Stumm and Sulzberger, 1992). Surface catalytic effects may include *direct* and *indirect* sorption-induced catalysis. In the environment, pH has been shown to have a strong influence on these sorption capacities of Fe minerals in general (Fowle and Konhauser, 2011). Considering the point of zero charge (PZC) of vivianite, which is with 3.3 below the lowest tested pH in our experiments, the mineral surface is positively charged under our experimental conditions (Luna-Zaragoza et al., 2009). Hence the pH range tested here will not affect the surface charge, and $NO_2^-$ sorption onto mineral surfaces and corresponding heterogeneous reactions are possible. In contrast, cell surfaces are considered to be negatively charged (Wilson et al., 2001) and therefore might induce different effects than mineral surfaces. The charge of the cell surface most likely remained negative even after autoclaving (see e.g. Halder et al., 2015). Our results imply that the systematics of chemodenitrification are strongly dependent on the surface provided and that, depending on the availability and quality of catalytic surfaces, Fe coupled chemodenitrification may be a single-step reaction (between $NO_2^-$ and Fe) or may occur in multiple steps (reaction between Fe and $NO_2$, as well as Fe and NO). As a consequence, the nature of surface catalysis would likely have a strong impact on the $N_2O$ yield per mole $NO_2^-$ reduced to NO. Since NO has been demonstrated to have a rather exceptional affinity towards $Fe^{2+}$ and $Fe^{3+}$ centres resulting in the formation of $Fe^{x+}(NO)_n$ nitrosyls and thus triggering an enhancement of the $N_2O$ decomposition rate (e.g. Rivallan et al., 2009). It remains unclear to what extent, and why, the quality of the catalytic surfaces plays a role. Particularly in the presence of organics and/or dead bacterial cells, which are known to have a high affinity to bind metal ions ( e.g. $Ni^{2+}$, $Cu^{2+}$ or $Zn^{2+}$), either directly or by forming surface complexes with hydroxyl groups (Fowle and Konhauser, 2011), a surface-catalysis-induced reaction can be expected. Besides acting as a catalyst via a reactive surface, the dead biomass might also have directly triggered the reaction. For example, non-enzymatic NO formation was studied and modelled by Zweier et al. (1999), suggesting that at concentrations between 100 and 1000 μM, abiotic $NO_2^-$ disproportionation and thus NO formation at circumneutral pH in organic tissue is



still possible (Zweier et al., 1999). Furthermore, autoclaving might have ruptured cell walls and released organic compounds.
In the presence of phenolic compounds, humic substances, and other organic compounds, $NO_2^-$ has been shown to form NO
via self-decomposition (Nelson and Bremner, 1969; Stevenson et al., 1970; Tiso and Schechter, 2015). Whether this may have
been the case also in our experiments remains unclear, since we did not conduct experiments containing only DB and $NO_2^-$.
Another possible consideration is the presence of extracellular polymeric substances (EPS), which should also be tested in
future studies. Liu et al., (2018) investigated nitrate-dependent Fe(II) oxidation with *Acidovorax* sp. strain BoFeN1, showing
that *c*-cytochromes were present in EPS secreted which could indeed act as electron shuttling agents involved in electron
transfer supporting chemolithotrophic growth. Since *S. oneidensis*, our model organisms used as DB supply, is known to
produce large amounts of EPS, harbouring *c*-cytochromes (Dai et al., 2016; Liu et al., 2012; White et al., 2016), a potential
impact of EPS on the reaction between $NO_2^-$ and Fe(II) needs to be considered. However, possible cytochromes present in the
EPS most likely lost their activity due to protein denaturation during autoclaving (Liu & Konermann, 2009; Tanford, 1970).
Nevertheless, EPS is still present and can act as a catalysing agent to the abiotic reaction mechanism (Klueglein et al., 2014;
Nordhoff et al., 2017).
Fe(II)$_{total}$ oxidation via $NO_2^-$ has also been observed in the mineral-only setups, but to a lower extent. Hence, the vivianite
mineral surfaces themselves seem to catalyse the abiotic reaction between $NO_2^-$ and Fe(II)/ $Fe^{2+}$ (in parts, the stimulation of
Fe-dependent nitrite reduction may also be attributed vivianite dissolution providing ample Fe(II) substrate). Previous studies
reported on mineral-enhanced chemodenitrification (Dhakal et al., 2013; Grabb et al., 2017; Klueglein & Kappler, 2013;
Rakshit et al., 2008), and the catalytic effect may be due to $NO_2^-$ adsorption onto the minerals surface possibly facilitating a
direct electron transfer. Similar findings have been reported previously on Fe(II) oxidation promoted by electron transfer
during adsorption onto a Fe(III) minerals surface (Gorski and Scherer, 2011; Piasecki et al., 2019). $OH^-$ adsorption is probably
enabled by the minerals positive surface charge at pH >6, resulting in a limited reactive surface availability. Complexation of
dissolved $Fe^{2+}$, which is provided by mineral dissolution, by $OH^-$ groups would thus result in a lower overall $NO_2^-$ reduction
rate compared to the DB-amended setups. Nevertheless, the NO formed by the initial $NO_2^-$ reduction could, at still elevated
$Fe^{2+}$ levels, proceed until both dissolved and adsorbed Fe(II) is quantitatively oxidized to surface-bound Fe(III) (Kampschreur
et al., 2011). This would ultimately lead to similar Fe(II)$_{total}$ oxidation and $N_2O$ production (and thus higher $N_2O$ yields) as in
the DB amended experiment and thus explain the similar results.

### 4.3. Mineral alteration during Fe-coupled chemodenitrification

We used $^{57}Fe$ Mössbauer spectroscopy in order to determine, whether the catalytic effects that enhanced chemodenitrification
with $Fe^{2+}$ also modulated mineral formation. In both setups, addition of Fe(II)Cl$_2$ to the 22 mM bicarbonate buffered medium
led to the formation of vivianite, an Fe(II)-phosphate. Shortly after the addition of $Fe^{2+}_{aq}$, the mineral phase in both setups was
dominated by Fe(II), but a small fraction of Fe(III) was also present. Initial fractions of Fe(III) were similar in both the mineral-
only and DB-amended experiments (9.9% and 7.4%, respectively) and, if not an artefact of Mössbauer sample handling, might
therefore have stimulated Fe(II) adsorption and oxidation (Gorski and Scherer, 2011; Piasecki et al., 2019). The reduction of

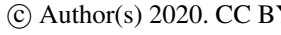



$NO_2^-$ was accompanied by a marked increase of Fe(III), likely in the form of short-range ordered ferrihydrite or lepidocrocite.
Thus, the Fe(III) phase detected at day 0 most likely formed immediately after $NO_2^-$ addition. This is supported by prior studies,
which demonstrated the initiation of Fe(II) oxidation with $NO_2^-$ within a short period of time (Jamieson et al., 2018; Jones et
al., 2015). At the end of the DB experiment at pH 6.89, oxidized Fe(III) (most likely in the form of poorly ordered ferrihydrite)
contributed 48.7% to the total Fe phases, with vivianite accounting for the remaining spectral area. Unfortunately, we are
unable to compare the results of the DB-amended precipitates at the end of the experiment to the mineral-only setup, since the
sample was lost. In contrast to our observations, other studies conducted in the presence of organics have identified goethite
as the main Fe(III) phase during the abiotic reaction between Fe(II) and $NO_2^-$ (Chen et al., 2018; Liu et al., 2018). In NDFeO
experiments, the formation of lepidocrocite, goethite, hematite and to some extent, magnetite has been reported (e.g. Klueglein
et al., 2014; Liu et al., 2018; Miot et al., 2015). In contrast, minerals obtained from the enrichment culture KS were mostly
vivianite and ferrihydrite, which is, however, attributed to the fact that for the cultivation of the KS culture a high-phosphate
medium is used (Nordhoff et al., 2017). In the abiotic experiments (10 mM Fe(II) and 10 mM $NO_2^-$) presented by Jones et al.,
(2015), the formation of lepidocrocite, goethite and two-line ferrihydrite were observed after 6 to 48 hrs. In the experiments
presented here, besides a short-range ordered Fe(III) phase, likely ferrihydrite, no other mineral phases could be identified
after 28 days.
Iron analysis also indicates that the oxidation of the Fe(II)$_{total}$ went to completion at pH 5.8 whereas at pH 6.8, 52.3% of the
Fe(II)$_{total}$ remained at the end of the incubation experiment, resulting in the formation of a poorly-ordered ferrihydrite.
Unfortunately, we did not measure the zeta potential of the starting solutions, which would probably help to explain the
differences detected. We note that, although $^{57}Fe$ Mössbauer spectroscopy was used to measure the Fe(II)/Fe(III) in the
precipitates, the reported Fe(II)$_{total}$ concentrations reflect the total Fe(II), i.e., of both the dissolved pellet (structurally-bound
or adsorbed) and the aqueous $Fe^{2+}$ in the supernatant measured by Ferrozine. The results obtained by Mössbauer analysis (50%
Fe(II) remaining) seem to contradict the ferrozine assay (<10% remaining) (see Table 1 and 2). The presence of ferrous Fe,
either as structurally-bound Fe(II) or adsorbed $Fe^{2+}$ does indeed play a crucial role with regards to the reaction dynamics
occurring at the mineral surfaces, particularly if we assume that N-reactive species are also still present (Rivallan et al., 2009).
In addition, the initially formed Fe(III) phase might also induce another feedback to the N and even the Fe cycle since Fe(III)
minerals are also highly reactive (Grabb et al., 2017; Jones et al., 2015). Mineral structure and thus Fe(II) location within the
lattice can influence the overall Fe accessibility, the binding site at the mineral surface and thus overall reactivity (Cornell and
Schwertmann, 2003; Luan et al., 2015; Schaefer, 2010). If the initial formation of Fe(III), however, enhanced the reaction
between $NO_2^-$ and Fe(II), similar results in both setups should have been observed, which this was not the case since $NO_2^-$
reduction patterns in the mineral-only experiments were much lower. This also indicates again, that the presence of DB indeed
contributed greatly to the reaction in the DB experiments. Furthermore, results obtained from Mössbauer analysis are the only
results supporting a pH-dependent effect: At pH 5.78 and in the presence of DB, all vivianite was fully transformed into a
short-range ordered Fe(III) phase whereas at pH 6.89, vivianite remained a major component. This presence of vivianite also
indicates that no further Fe(II) oxidation occurred even though $NO_2^-$ reduction was incomplete. The incomplete reduction of





$NO_2^-$ in turn suggests that further Fe(II) oxidation was limited due to blocked or deactivated reaction sites on mineral surfaces.
Also, considering that at pH 5.8 and in the presence of DB, the initial $NO_2^-$ concentrations were higher but the overall reaction
dynamics were quite similar to the other reaction conditions, the concentration dependency of the reaction between $NO_2^-$ and
Fe(II) is again supported.
**4.4. Nitrite and $N_2O$ N and O isotope dynamics during chemodenitrification**
In the presence of only vivianite, a decrease in $\delta^{15}N$-$NO_2^-$ of ~3‰ was observed with the initial decrease in $NO_2^-$. Initial $\delta^{18}O$-
$NO_2^-$ values also reflect this drop of 3‰ during the first 3 days but level off and stabilize at 1‰ after 9 days. The initial decrease
in both $\delta^{15}N$ and $\delta^{18}O$ of $NO_2^-$ suggest apparent inverse isotope effects, which to the best of our knowledge have never been
observed during chemodenitrification, and have only been reported for enzymatic $NO_2^-$ oxidation (Casciotti, 2009). Since
biological $NO_2^-$ oxidation can be ruled out (no $NO_3^-$ produced, no microbes), the decrease in $\delta^{15}N$-$NO_2^-$, though subtle, could
indicate that either heavy isotopes are incorporated in the products formed (i.e. NO, $N_2O$), at least at the beginning of the
incubation period. Normally, the heavier isotopes build compounds with molecules of higher stability (Elsner, 2010; Fry, 2006;
Ostrom & Ostrom, 2011). This is particularly true for the formation of some minerals or highly stable molecules that are
formed under mineral-only conditions, where processes can reach an isotopic equilibrium (He et al., 2016; Hunkeler & Elsner,
2009; Li et al., 2011; Ostrom & Ostrom, 2011). However, in the system presented here, N incorporation into mineral phases
can be excluded, hence another process must favour the heavy N-atoms. Since this initial drop in $\delta^{15}N$ was also observed in
the DB-amended experiments, a possible explanation might be that the isotope values here reflect the sorption or complexation
mechanism of $NO_2^-$ onto the reactive surfaces. In contrast $\delta^{18}O$-$NO_2^-$ values, after the initial decrease, did not change greatly
with decreasing $NO_2^-$ concentrations. The stabilization of the $\delta^{18}O$-$NO_2^-$ towards the end of the experiment most likely reflects
the oxygen isotope equilibration between $\delta^{18}O$-$NO_2^-$ and the $\delta^{18}O$ of the water in the medium. Temporal $\delta^{18}O$-$NO_2^-$ dynamics
did not change greatly between the different pH treatments, and in all cases the final $\delta^{18}O$-$NO_2^-$ ranged between 0.5 and 1‰.
The kinetics of abiotic O-atom exchange is a function of temperature and pH. At near neutral pH, at room temperature, one
can expect $NO_2^-$ to be fully equilibrated after two to three days (Casciotti et al., 2007). At higher pH, the first order rate
constants for the equilibration with water are lower (Buchwald and Casciotti, 2013), but equilibrium conditions should have
been reached well within the incubation period. Indeed, the final $\delta^{18}O$-$NO_2^-$ was consistent with an equilibrium O isotope effect
between $NO_2^-$ and $H_2O$ with a $\delta^{18}O$ of ~ -11.5‰ (Buchwald and Casciotti, 2013). With regards to $\delta^{15}N$-$NO_2^-$ values of the DB-
amended experiments, a similar behaviour is found within the first 3 days (i.e., decrease in $\delta^{15}N$), followed by a clear increase
in $\delta^{15}N$-$NO_2^-$ of ~10‰. While it is difficult to explain the initial decrease in $\delta^{15}N$-$NO_2^-$ (a feature that was not observed in other
chemodenitrification experiments (i.e. Grabb et al., 2017; Jones et al., 2015), the subsequent increase in $\delta^{15}N$ can be attributed
to normal isotopic fractionation associated with chemodenitrification and an N isotope effect (-9‰) that is consistent with
those previously reported on Rayleigh-type N and O isotope kinetics during chemodenitrification with Fe(III)-bearing minerals
such as nontronite and green rust (Grabb et al., 2017). In contrast, $\delta^{18}O$-$NO_2^-$ values initially decrease as in the abiotic
experiment but then level off faster reaching final values of ~1‰, again most likely explained by O atom isotope exchange





pulling the $\delta^{18}$O-NO$_2^-$ values towards the O-isotope equilibrium value. This value is given by the $\delta^{18}$O$_{H2O}$ + $^{18}\varepsilon_{eq,NO2-}$, whereas
the latter is defined as the equilibrium isotope effect between NO$_2^-$ and H$_2$O and has been shown to yield values of roughly
+13‰ (Casciotti et al., 2007). Overall, it seems that the non-linear behaviour of the NO$_2^-$ in the O isotope Rayleigh plot is most
likely due to the combined effects of kinetic O isotope fractionation during NO$_2^-$ reduction, and O atom exchange between
NO$_2^-$ and H$_2$O.
NO$_2^-$ N and O isotope trends observed under the DB-amended conditions (in which a large portion of the NO$_2^-$ pool was
consumed), somewhat contradict prior reports of chemodenitrification exhibiting a clear increase in both $\delta^{15}$N and $\delta^{18}$O-NO$_2^-$,
with N isotope enrichment factors for NO$_2^-$ reduction between -12.9 and -18.1‰ and an O isotope effect of -9.8‰ (Jones et
al., 2015). Consistent with our data, however, they also observed that, at least in abiotic experiments where NO$_2^-$ consumption
is rather sluggish due to Fe$^{2+}$ limitation (as a result of either oxidation or simply occlusion), O-isotope exchange isotope effects
mask the effects of kinetic O isotope fractionation. While we cannot say at this point what exactly governs the combined NO$_2^-$
N vs. O isotope trends in the two different experimental conditions, we observed that the two processes (water isotope
equilibrium and KIE) competing with each other lead to different net dual isotope effects. Our data cannot resolve whether
these observations reflect fundamental differences or simply changes in the relative proportion of the competing processes.
Nevertheless, our observations may still be diagnostic for chemodenitrification catalysed by a mineral surface on the one hand,
and Fe-coupled chemodenitrification that involves catalytic effects by dead bacterial cells on the other. The mineral catalyst
evidently plays an important role with regards to chemodenitrification kinetics, reaction conditions, surface complexation or
contact time between the NO$_2^-$ substrate and the mineral phase (Samarkin et al., 2010), and in turn the combined
kinetic/equilibrium N and O isotope effects.
The $\Delta^{15}$N values ($\Delta^{15}$N= $\delta^{15}$N$_{nitrite}$ - $\delta^{15}$N$_2$O$^{bulk}$) presented in Table 3 were obtained by subtracting the average $\delta^{15}$N$^{bulk}$ value of
N$_2$O (abiotic -46.5 ±0.2‰; dead biomass -49.4 ±1.0‰) across all pH and throughout the experiment from the average of the
initial $\delta^{15}$N$_{nitrite}$ value. These values can provide insight on reaction kinetics between NO$_2^-$, NO, and N$_2$O (Jones et al., 2015).
In both setups there is an offset between the NO$_2^-$ and N$_2$O $\delta^{15}$N, which is clearly higher than what would be expected based
on the NO$_2^-$ reduction NO$_2^-$ isotope effect of <10‰. Following the argumentation of Jones et al. (2015), who reported a similar
N isotopic offset between NO$_2^-$ and N$_2$O of 27.0 ±4.5‰, this could be indicative for a heavy N accumulating in a forming NO
pool, whereas $^{14}$N is preferentially reacting to N$_2$O or N$_2$, respectively. This might even be supported by the rather low $\delta^{15}$N$^{bulk}$
values detected for N$_2$O in both setups.










**Table 3: Comparison of the isotope values obtained during dead biomass versus the abiotic experiments. T0 values represent means calculated by summarizing results across all pH ± standard error. $\delta^{15}N$ and $\delta^{18}O$ values were calculated using $\bar{x}_{t0} - \bar{x}_{tend}$. Isotope fractionation was calculated is based on the slope between the lowest initial value (here at $t_1$) and $t_{end}$ for all pH. $\Delta^{15}N$ (= $\delta^{15}N_{nitrite}$ - $\delta^{15}N_2O^{bulk}$) was calculated for the end of the experiment.**

|  | Dead Biomass | Abiotic |
|---|---|---|
| $\delta^{15}N_{nitrite}(t_0\text{-}t_{end})$ | ↓5.99 ±0.65‰ | ↓5.93 ±0.73‰ |
| $\delta^{18}O_{nitrite}(t_0\text{-}t_{end})$ | ↓1.75 ±0.23‰ | ↓1.15 ±0.18‰ |
| $^{15}\varepsilon_{nitrite}$ | -10.36 ‰[#] | - |
| $^{18}\varepsilon_{nitrite}$ | -0.51‰[#] | - |
| SP | 1.17 ±1.2‰ | 5.99 ±0.84‰ |
| $\delta^{15}N^{\alpha}$ | -51.84 ±0.1‰ | -43.53 ±0.16‰ |
| $\delta^{15}N^{bulk}$ | -49.38 ±1.01‰ | -46.48 ±2.1‰ |
| $\Delta^{15}N$ | 23.2‰ | 27.85‰ |

[#] n=4 ($t_1$ to $t_{end}$); - concentrations in abiotic experiment fluctuate and show only minor decrease, hence $^{15}\varepsilon$ and $^{18}\varepsilon$ could not be calculated.

While our results clearly showed that $N_2O$ accumulates over the course of the reaction, it remains unclear, which additional end products are present at the final stage of the experiment. If NO accumulates (instead of following the reaction cascade further), the substrate-product relationship between the $\delta^{15}N\text{-}NO_2^-$ and $\delta^{15}N\text{-}N_2O$ values that would be expected in a closed system is perturbed, leading to significantly higher $\Delta^{15}N$ than predicted by the $\delta^{15}N\text{-}NO_2^-$ trend. Hence, the calculated $\Delta^{15}N$ of the mineral-only treatment (27.9‰) is only slightly higher than that of the DB experiment (23.2‰), and would therefore suggest that despite the differences in chemodenitrification kinetics (i.e., different $NO_2^-$ reduction rates and extent), the NO pool formed is enriched in heavy N in both treatments, respectively. Alternatively, fractional reduction of the produced $N_2O$ to $N_2$ may also affect the $\Delta^{15}N$ since it would presumably increase the $\delta^{15}N\text{-}N_2O$ and thereby raise the low $\delta^{15}N\text{-}N_2O$ closer to the starting $\delta^{15}N\text{-}NO_2^-$. Abiotic decomposition of $N_2O$ to $N_2$ in the presence of Fe-bearing zeolites has been investigated previously (Rivallan et al., 2009), however, it remains unclear if this process could also occur here. Fractional $N_2O$ reduction is also not explicitly indicated by the SP values, which would reflect an increase with $N_2O$ reduction (Ostrom et al., 2007; Winther et al., 2018). The SP values in both mineral-only and DB-amended experiments were, with some exceptions, relatively low (6.0 ± 0.8‰; 1.7 ± 1.2‰; Fig. 6). In fact, SP values observed during the course of our experiments are significantly lower compared to SP values reported in other studies on Fe-oxide-mineral associated chemodenitrification (e.g., ~16‰; Jones et al. (2015); 26.5‰; Grabb et al. 2017), or during the abiotic $N_2O$ production during the reaction of Fe and a $NH_2OH/NO_2^-$ mixture (34‰; Heil et al. 2014). While the variety of different SP values for chemodenitrification-derived $N_2O$ suggests different reaction conditions and catalytic effects, our SP data seem to imply that the mineral catalyst plays only a minor role with regards to the isotopic composition of the $N_2O$ produced. However, since $N_2O$ concentrations, even if minor, are increasing towards the end of the experiments, production and possible decomposition as well as ongoing sorption mechanisms might





also serve as possible explanation leading to these rather low SP values. $N_2O$ SP values have been used as valuable tracer for
microbial $N_2O$ production (Ostrom & Ostrom, 2012). Based on pure culture studies (Ostrom et al., 2007; Winther et al., 2018;
Wunderlin et al., 2013) and investigations in natural environments (Wenk et al., 2016) a SP range of -10 to 0‰ is considered
to be characteristic for denitrification or nitrifier denitrification (Sutka et al., 2006; Toyoda et al., 2005), whereas higher values
are usually attributed to nitrification or fungal denitrification (Ostrom & Ostrom, 2012; Wankel et al., 2017; Well & Flessa,
2009). The SP values reported here (0 to 10‰) fall well within the range of biological $N_2O$ production, explicitly denitrification
and soil derived denitrification (2.3 to 16‰) (Ostrom & Ostrom, 2012), rendering the separation between chemodenitrification
and microbial denitrification based on $N_2O$ isotope measurements difficult, if not impossible.
In summary, the N and O isotope systematics of chemodenitrification are multifaceted, depending on the environmental
conditions, reaction partners provided, and/or the speciation of precipitated mineral phases. The systematics observed here are
clearly not entirely governed by normal kinetic isotope fractionation only, as has also been observed in previous work. Grabb
et al. (2017) demonstrated that there is a relationship between reaction rate and kinetic $NO_2^-$ N and O isotope effects, with
faster reaction leading to lower $^{15}\varepsilon$ and $^{18}\varepsilon$. Again, changes in the expression and even in the direction of the isotope effects in
the $NO_2^-$ pool suggest that multiple processes, including equilibrium isotope exchange (at least with regards to the $\delta^{18}O$- $NO_2^-$
), are contributing to the net N and O isotope fractionation regulated by the experimental conditions and reaction rates. As
pointed out by Grabb et al. (2017), and as supported by our comparative study with pure abiotic mineral phases and with added
dead biomass, the accessibility of Fe(II) to the reaction may be a key factor regarding the degree of N and O isotope
fractionation expressed, particularly if complexation limits the reactive sites of the mineral. The conditions that, at least
transiently, lead to the apparent inverse N and O isotope fractionation observed here for chemodenitrification requires
particular attention by future work. At this point, we can only speculate about potential mechanisms, which are indicated in
the conceptual illustration (Figure 8). As chemodenitrification seems to be catalysed by reactive surfaces of Fe(II)/Fe(III)-
minerals and/or organics (including cells), sorption onto these surfaces might play a crucial role in the fractionation of N and
O isotopes. For example, during the catalytic hydrogenation of $CO_2$ on Fe and Co catalysts a subtle depletion (ca. 4‰) in
$^{13}CO_2$ at progressed conversion to methane has been explained by the precipitation of a $^{13}C$-enriched carbon intermediate (e.g.,
CO-graphite) on the catalyst surface (Taran et al., 2010). We are fully aware that it is difficult to compare our system with
Fischer-Tropsch synthesis of methane occurring at high temperature and pressure. Yet given the indirect evidence for NO
accumulation in our experiments, it may well be that preferential chemisorption/complexation of "heavy" intermediate NO
occurs, which may lead to transient $^{15}N$-depletion in the reactant $NO_2^-$ pool. Considering that the $N_2O$ concentrations measured
in our experiments were comparatively low and that $\delta^{15}N^{bulk}$ $N_2O$ values did not noticeably change throughout the experiments,
formation of $N_2$ via abiotic interactions between $NO_2^-$ and NO may also be involved (Doane, 2017; Phillips et al., 2016). Hence,
$N_2O$ is meddling with the reaction dynamics either as an intermediate or as a side product, and can thereby influence the overall
N and O isotope dynamics.






Figure 8: Conceptual figure depicting the proposed reaction mechanisms and feedbacks between the different N species during
chemodenitrification induced by the presence of a mineral surface (lower left corner) or (dead) biomass (upper right corner).
Adsorption of $Fe^{2+}$ (directly or via complexation by $OH^-$) as well as $NO_2^-$ could catalyse a direct reaction between both. In addition,
$NO_2^-$ adsorption onto the Fe(II) mineral might also induce disproportionation, leading to $NO_x$ formation. These formed
intermediates, although transitory, may impact the overall reaction dynamics by e.g. complex formation (i.e. [NO--$Fe^{2+}$]) or direct
Fe(II) oxidation. The produced Fe(III) might induce another feedback loop (autocatalysis) resulting in further Fe(II) oxidation.
Similar processes are possibly induced by the presence of (dead) biomass. Adsorption and complexation of either $NO_2^-$ and $Fe^{2+}$
would enhance the reaction between both. In addition, the presence of organic acids would decrease the pH locally and thereby
promote and accelerate $NO_2^-$ disproportionation and thus additionally enhance Fe(II) oxidation. Our results suggest that $NO_2^-$
reduction results in an KIE, which should influence the isotopic composition of NO. $N_2O$ here is an intermediate, the isotopic
composition of which is mainly influenced by an EIE between NO and $N_2O$. The low $N_2O$ yields as well as the $N_2O$ isotopic results
(bulk, SP) clearly suggests that $N_2$ is produced abiotically.






## 5. Conclusions and outlook


In the absence of any clear (genetic) evidence for enzymatic NDFeO from cultures (e.g. *Acidovorax* sp. strain BoFeN1),
heterotrophic denitrification/$NO_3^-$ reduction coupled to abiotic oxidation of Fe(II) with the $NO_2^-$ has been presented as the most
reasonable explanation for NDFeO. Here we investigated the second, abiotic step, clearly demonstrating that Fe-associated
abiotic $NO_2^-$ reduction can be catalysed by mineral and organic phases under environmentally relevant conditions, as found
for example in soils and aquifers. Our results confirm that reactive surfaces play a major role with regards to the reaction
between $NO_2^-$ and Fe(II) and that surface-catalysed chemodenitrification appears to not only contribute to the production of
the greenhouse gas $N_2O$ in environments hosting active cycling of Fe and N, but also to an abiotic production of $N_2$. In order
to understand the mechanistic details of Fe-coupled chemodenitrification, natural-abundance measurements of reactive-N
isotope ratios may help distinguish between abiotic and biotic reactions during NDFeO. Our results, however, indicate that the
potential of coupled N and O isotope measurements to determine the relative importance of Fe-induced N-transformations in
natural environments is somewhat limited. Considering, for example, the apparent inverse N isotope effect in the mineral-only
experiments, our studies show that the $NO_2^-$ N vs. O isotope systematics seem to contrast distinctly between biotic and abiotic
$NO_2^-$ reduction, potentially permitting the disentanglement of the biotic versus abiotic processes. $N_2O$ SP values seem to be
less diagnostic with regards to discriminating between chemodenitrification-derived $N_2O$ and $N_2O$ that is produced during
microbial $NO_2^-$ reduction. Our results suggest that both the reaction between Fe(II) and reactive N species, as well as the
resulting isotope effects, are dependent on the reactive surfaces available. The presence of organic material seems to enhance
$NO_2^-$ reduction and, to a lesser extent also $N_2O$ production, leading to the enrichment in $^{15}N$ in the residual $NO_2^-$, as predicted
by Rayleigh-type kinetic N isotope fractionation. In the presence of only Fe(II) minerals, $NO_2^-$ reduction rates are significantly
lower, and net N and O isotope effects are not governed by kinetic isotope fractionation only, but also by isotope equilibrium
fractionation during exchange with the ambient mineral phase and/or the ambient water (in the case of O isotopes). While $N_2O$
production was significant, the $N_2O$ yields were below 5%, suggesting that a significant fraction of the $NO_2^-$ reduced is at least
transiently transformed to NO and possibly $N_2$. This transient pool of NO possibly stands in quasi-equilibrium with other
intermediates (i.e. HNO, $NO_2(g)$) or complexes (i.e. Fe-NO), and may thereby impact the overall reaction kinetics as well.
We speculate that the transient accumulation of NO represents an important constraint both on overall reaction kinetics as well
as on the $N_2O$ isotopic signature (or $\Delta^{15}N$), an aspect that should be verified in future work. Such work may include the
quantification of $N_2$ (and its N isotopic composition), which will help to assess to what extent (i) Fe-mineral surface-induced
chemodenitrification leads to the formation of a transient pool of NO and is driven by the catalytically induced abiotic reaction
between Fe(II) and $NO_2^-$, or if (ii) NO is actually the main oxidizing agent of Fe(II).
Our data revealed further complexity with regards to N and O isotope effects during Fe-coupled chemodenitrification than
previously reported. We argue that its isotopic imprint depends on the substrate concentration, the presence of reactive surfaces
or other catalysts, the mechanisms induced by these catalysts (e.g. surface complexation), and putatively on the intermediates
as well as on the product present at the end of the experiments. The multifaceted control on coupled N and O isotope




systematics in reactive N species may explain the discrepancies observed between our and previous work (e.g., with regards to $^{15}\varepsilon$:$^{18}\varepsilon$ ratios; Grabb et al. 2017). Clearly, one has to be realistic with regards to using $NO_2^-$ and/or $N_2O$ N and O isotope measurements to provide constraints on the relative importance of chemodenitrification under natural conditions. Yet, at this point, there is only a very limited number of studies on the isotope effects of chemodenitrification, and with the results presented here, we expand the body of work that aims at using stable isotope measurements to assess the occurrence of chemodenitrification in denitrifying environments. More work on the controls of stable isotope systematics of chemodenitrification, in particular on the role of reactive, and potentially cryptic, intermediate N species, and of O isotope exchange, will improve our ability to more quantitatively trace Fe-coupled nitrite reduction and $N_2O$ production in natural Fe-rich soil or sedimentary environments.

**Data availability**

Data can be accessed upon request to the corresponding author.

**Author contributions**

AAK initiated the project. MFL and AAK supervised the project. ANV designed and conducted all experiments. Isotope measurements as well as data analysis were performed by ANV under the supervision of MFL. JMB conducted Mössbauer measurements and data analysis. PAN supervised and performed all $N_2O$ concentration determination measurements. ANV, SDW and MFL interpreted the data and prepared the paper with inputs from all other co-authors.

**Competing interests**

The authors declare that they have no conflict of interest.

**Acknowledgements**

Special thanks go to Karen L. Casciotti (Stanford University) for helping with the correction of the $N_2O$ isotope data. Thanks to Cindy-Louise Lockwood for corrections and comments on earlier versions of the manuscript, and to Viola Warter, Elizabeth Tomaszewski for fruitful discussions on abiotic chemistry and mineral reactions. Markus Maisch is thanked for his help with the preparation of the Mössbauer samples.



## Funding

This research was supported by the Deutsche Forschungsgemeinschaft - DFG (Grants GRK 1708 Molecular principles of bacterial survival strategies), and through funds from the University of Basel, Switzerland.

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
