# Peer review of "Impact of reactive surfaces on the abiotic reaction between nitrite and ferrous iron and associated nitrogen and oxygen isotope dynamics"

_Biogeosciences, 2020_

## Referee Comment (RC1) · Anonymous Referee #1 · 12 Jun 2020

This study is based on laboratory experiments and aims to trace the NDFeO. The work is colossal with a large amount of data obtained. I regret that some of the data are not presented or discussed (see my comments). The paper is very well written but I am stuck on the fact that the decrease in nitrite concentrations is accompanied by (i) a depletion and then an enrichment of d15N, (ii) that the 18O of the residual nitrite in the end varies very little. There is no mass balance, which makes it difficult to monitor the extent of the processes. If the nitrite is transformed into N2O and this accumulates, it is very easy to calculate what its $\delta$15N and $\delta$18O should be compared with the measured

values. My more detailed comments below :

L39-40: I'm surprised there are no older references to the role of iron. General experiment setup section : The conditions of the experiment are anoxia and the addition of iron and nitrogen in the form of nitrite. Under these conditions, in the environment, it is conceivable that dissimilative reduction of nitrite to ammonium may occur. Of course under perfect abiotic conditions DNRA should not occur. Did the authors measure ammonium concentrations throughout the experiment to ensure that no other processes than the one under study were taking place?

L120-121 : How long does it take from incubation to the measurement of concentrations and isotopes? Light is a factor that can generate abiotic reactions, which in turn can generate isotope fractionation. What about it?

L179-180 : Two nitrite isotope standards have been used. What are the values of these standards? Do these values include those of the samples measured in this study? What is the analytical precision of the method (preparation + intrinsic analysis) for the determination of the isotopic composition of nitrite (15N and 18O)?

L285-291 : Rayleigh conditions allow the isotope fractionation factor to be easily determined by looking at the slope of the line on a representation ln C/C0 as a function of d15N, but not C (with C the concentration at time t and C0 the initial concentration). This paragraph is not clear to me. Moreover, doesn't the fact that there is first a decrease of 15N, i.e. an inverse isotopic fractionation, with a decrease of the amount of heavy isotope in the residual substrate, and then an enrichment, mean that several processes could take place and that process 1 takes place at the beginning of the experiment with a higher rate than the second process which either starts at the beginning of the experiment or when process 1 is completed? Very concretely, the trend line is calculated on the points starting from the lowest d15N values? I think it would be necessary to clarify this part.

L296-302: Is it not possible to envisage that the variations in 18O are due solely to an

exchange between the oxygen of the nitrite and the oxygen of the water? By the way, what is the isotopic composition of water? Is it constant during the experiment?

L309-313: The authors have done a significant analytical work. Why not show the variations in N2O concentration as a function of nitrite concentrations. Before any interpretation with isotopes or isotopologists, it seems to me useful and necessary to work on the concentrations and in particular to make mass balances.

L314-315: The authors do not discuss the very negative SP value, which is very distinct from the other points. Is this an analytical problem?

L326 : There is no figure S6. But mentioned in S5 section figure 3.

L484-486 : Large variations of $\delta$15N are not associated with variations of $\delta$18O. While these are measurements made on the residual substrate. The drop in 18O at the beginning of the experiment is more likely due to an isotopic exchange with the oxygen in the water than evidence of a process.

L531-538 : It might be interesting to look at $\delta$18O variations of N2O during the experiment. And see if it correlates with that of nitrite. This would also be an opportunity to confirm or deny whether there is an isotope exchange between the oxygen in the nitrite and the oxygen in the water.

L551-552: if N2O is considered to accumulate, it can be considered to be the accumulated product in the case of a Rayleigh distillation. In this case, and taking into account the isotope fractionation associated with nitrite reduction, it is easy to calculate what the expected 15N and 18O of the N2O produced. It would then be interesting to compare the measured values with the expected values.

---

## Referee Comment (RC2) · Anonymous Referee #2 · 20 Jun 2020

This manuscript investigates in detail chemodenitrification mechanisms and presents an attempt of using nitrite isotopic analyses to increase processes understanding and ability to distinguish them. This is an important and interesting study, manuscript is in general well organised and reader-friendly. I have two main points to be clarified: 1) the SP values of N2O - some of your values are extremely low - down to -120 permil, which is absolutely implausible value - at least so far no known process could explain these values. You ignore them in your results description and discussion. How reliable are these results? Why to show them and not discuss them? I think they must be wrong, probably due to some problems with measurements. But this makes me question all the SP values of N2O - if we see such a large bias for some samples, are we sure other results are true? Especially when your SP data are significantly lower compared to previously published values. Maybe all your values are underestimated? If you cannot explain this I would suggest to remove these data from the manuscript - this will also do without these data and will not bring confusion for the further studies. 2) abiotic N2O reduction to N2 - in some places you say it is unclear if this is possible, in other you say results suggest this occurs (I have also indicated this in the attached file). There is no clear point about this process in the manuscript.

Please find the further specific comments in the attached pdf file.

Please also note the supplement to this comment:
https://www.biogeosciences-discuss.net/bg-2020-73/bg-2020-73-RC2-supplement.pdf

**Supplement:**

[revised manuscript text omitted]

---

## Author Comment (AC1) · 3 Jul 2020

First, we wish to thank the reviewer for his/her valuable inputs and comments on our manuscript.

L39-40: I'm surprised there are no older references to the role of iron.

Reply: We agree that indeed there are many more references regarding the role of iron in the environment. However, our choice can be considered as "best of" selection, covering a whole suite of different aspects: We choose (1) Expert et al., 2012 since

they explicitly focus on the vital role of iron for all living organisms, its wide range of redox potentials and its catalytic role in various metabolic pathways; (2) Lovley et al., 1997, who reported on the importance of iron already in 1988, however, the publication chosen represents a nice "summary", focusing also on various reactions and thus its "remediative" capabilities. Obviously, we wanted to limit the number of references, but if the reviewer has a specific publication in mind, we will be happy to include it. Again, in light of the many publications on the importance of iron available, and since our manuscript is already very long, we simply decided to pick two references that support the statement/sentence.

General experiment setup section: The conditions of the experiment are anoxia and the addition of iron and nitrogen in the form of nitrite. Under these conditions, in the environment, it is conceivable that dissimilative reduction of nitrite to ammonium may occur. Of course under perfect abiotic conditions DNRA should not occur. Did the authors measure ammonium concentrations throughout the experiment to ensure that no other processes than the one under study were taking place?

Reply: As the reviewer stated, DNRA should not occur under abiotic conditions. Considering that the abiotic experiments were all performed under laboratory conditions, using a medium that contains already high amounts of ammonium (5.61 mM NH4Cl, see 2.1), ammonium concentrations were only checked sporadically for some setups. Since only (if at all) minor fluctuations were observed, no further efforts to determine ammonium concentrations were attempted.

L120-121: How long does it take from incubation to the measurement of concentrations and isotopes? Light is a factor that can generate abiotic reactions, which in turn can generate isotope fractionation. What about it?

Reply: Yes, light-induced reactions have to be considered. That was one reason why nitrite concentrations were measured via CFA immediately after the samples were taken (within one hour). After determining the nitrite concentrations, the azide method

was applied (within max. 2-3 hrs). Samples were kept inside the glovebox in coloured (dark brown or blue) Eppendorf tubes, whereas the latter were chosen to inhibit potential photocatalytic reactions. The azide-treated headspace vials were stored in card boxes at RT until measured. At this point, the sample is fixed (i.e., turned into N2O). Therefore, we are rather confident that neither light nor (possibly) temperature could have influenced the values. However, one could argue that the blue coloured Eppendorf tubes might not suffice, since they are indeed partly translucent. Since during one of the experiments blue and brown vials were used, and still, the concentration values within the nine replicates were very similar (see Figure 1 A and C, note error bars), we are confident that the rapid processing and precautions taken to avoid light-induced reactions did indeed suffice.

L179-180: Two nitrite isotope standards have been used. What are the values of these standards? Do these values include those of the samples measured in this study? What is the analytical precision of the method (preparation + intrinsic analysis) for the determination of the isotopic composition of nitrite (15N and 18O)?

Reply: Standard N-7373 has a $\delta$15N value of -79.6‰ and a $\delta$18O value of +4.5‰ In contrast, standard N-10219 has a $\delta$15N value of +2.8‰ and a $\delta$18O value of +88.5‰ Using both standards allowed for a reliable correction using standard bracketing: The standard $\delta$15N range included the $\delta$15N values obtained for our samples perfectly. The $\delta$18O values measured fell only slightly below (-0.5 to 2.5‰ the range given by the standards, so that corrections are reliable. Based on replicate measurements of laboratory standards and samples, the analytical precision for NO2- $\delta$15N and $\delta$18O analyses was ±0.4‰ and ±0.6‰ (1 SD), respectively.

L285-291: Rayleigh conditions allow the isotope fractionation factor to be easily determined by looking at the slope of the line on a representation ln C/C0 as a function of d15N, but not C (with C the concentration at time t and C0 the initial concentration). This paragraph is not clear to me. Moreover, doesn't the fact that there is first a decrease of 15N, i.e. an inverse isotopic fractionation, with a decrease of the amount

of heavy isotope in the residual substrate, and then an enrichment, mean that several processes could take place and that process 1 takes place at the beginning of the experiment with a higher rate than the second process which either starts at the beginning of the experiment or when process 1 is completed? Very concretely, the trend line is calculated on the points starting from the lowest d15N values? I think it would be necessary to clarify this part.

Reply: We agree, the title of the x-axis of Figure 5 might be misleading. Of course, the values of the x-axis represent the ln of the substrate fraction remaining (as mentioned in the caption below the figure). Hence, it is the ln(f) whereas f is C/C0. We will change the title of the x-axis to avoid future confusions. With regards to the second comment, i.e., that the data presented might simply reflect that two different processes are at work, we also agree. However, since it is hard to explain which processes might be at work and if this is indeed a clear inverse effect, we decided to calculate the isotope effect using the lowest $\delta$15N values observed (i.e. for the experimental period where we show a clear decline in nitrite concentration with a net increase in $\delta$15N). We will clarify that there is putative evidence for multiple processes occurring in the incubations, and that this has implications for the Rayleigh approach.

L296-302: Is it not possible to envisage that the variations in 18O are due solely to an exchange between the oxygen of the nitrite and the oxygen of the water? By the way, what is the isotopic composition of water? Is it constant during the experiment?

Reply: Unfortunately, the isotopic composition of the water was not measured, and we can only assume its $\delta$18O (the water used in Tübingen has a d18O of roughly 11‰. It is possible that the variations in $\delta$18O are partially attributable to oxygen atom exchange dynamics with the matrix (see e.g. L504-516). However, considering that the observed drop in $\delta$18O values in both experiments occurs more or less simultaneously with the drop in $\delta$15N might be indicative of other dynamics (e.g. sorption, complexation?).

L309-313: The authors have done a significant analytical work. Why not show the

variations in N2O concentration as a function of nitrite concentrations? Before any interpretation with isotopes or isotopologists, it seems to me useful and necessary to work on the concentrations and in particular to make mass balances.

Reply: The proposed graph could be added to the supplementary material. However, particularly for the mineral only setups, this way of visualizing the data does not help much (see Figure 1"N2O vs NO2- concentrations in (A) mineral plus dead biomass and (B) mineral only experiments). Also, for the main manuscript we had severe concerns with regards to its length. Therefore, we chose to present only graphs that really help to understand the main messages of this project. With regards to the mass balance: The initial objectives of this project included mass balance considerations since it was supposed to lay the ground for a following study on nitrate-dependent Fe(II) oxidation in selected microbial strains. Unfortunately, we did not have the capacities to also analyse the N2 samples, so a proper mass balance is unfortunately not possible.

L314-315: The authors do not discuss the very negative SP value, which is very distinct from the other points. Is this an analytical problem?

Reply: We assume that the reviewer is referring to the observed drops in SP values (-120 to -80‰, occurring at t1 for samples taken from the mineral + dead biomass setup at pH 6.2 and mineral only at pH 5.8. After another thorough check of the raw data, we have to admit that for those particular samples the peak areas of the data obtained via CF-IRMS were much higher (compared to standards), possibly causing an extreme linearity or contamination effect that is affecting the data. We re-checked the entire data set again and removed these outliers (see revised Figure 2 "Site Preference (SP; A, C) and $\delta15$Nbulk (B, D) values of N2O produced in experiments amended with mineral + dead biomass (red) and mineral-only (grey)"). The bulk of the data is not compromised, as we have good agreement between the standard and the sample peak areas.

L326: There is no figure S6. But mentioned in S5 section figure 3.

Reply: We thank the reviewer for pointing this out and apologize for the mistake. Figure
S5 mentioned in L322 actually corresponds to Figure S4 in the supplements, while S6 in L 326 refers to S5! We will change this in the re-submitted version of the MS.

L484-486: Large variations of $\delta15N$ are not associated with variations of $\delta18O$. While these are measurements made on the residual substrate. The drop in 18O at the beginning of the experiment is more likely due to an isotopic exchange with the oxygen in the water than evidence of a process. Reply: Whether the drop is solely caused by the O isotopic exchange or, maybe partially, by interactions with the mineral surface, is not really clear. The drop observed in $\delta18O$ occurs almost simultaneously with the e.g. the decrease in $\delta15N$ for the mineral + dead biomass experiment. This might be indicative of other processes playing indeed a certain role. However, as we tried to explain in L496ff in the original MS, we assume that the main effect is the oxygen exchange with the water of the medium, which simply takes time and thus results in "fluctuations" (especially for the mineral only experiments) until the entire system is equilibrated.

L531-538: It might be interesting to look at $\delta18O$ variations of N2O during the experiment. And see if it correlates with that of nitrite. This would also be an opportunity to confirm or deny whether there is an isotope exchange between the oxygen in the nitrite and the oxygen in the water. Reply: Indeed, using the $\delta18O$ variations of N2O might help to better understand the isotope exchange processes within the system. However, since N2O is definitely not the only product and possibly further reduced (resulting in a branching effect caused by the removed O atoms, which is further affecting the O dynamics within the system), this approach would be biased.

L551-552: if N2O is considered to accumulate, it can be considered to be the accumulated product in the case of a Rayleigh distillation. In this case, and taking into account the isotope fractionation associated with nitrite reduction, it is easy to calculate what the expected 15N and 18O of the N2O produced. It would then be interesting to compare the measured values with the expected values.

Reply: We agree that it is indeed possible to estimate the predicted value of $\delta15N$ by using the accumulated product equation. An epsilon value calculated from the $\delta15N$-NO2- data could be used to estimate the predicted $\delta15N$-N2O values, which would be different since N2O is clearly not the single product. However, for $\delta18O$ this approach would not work due to the branching effect occurring during reduction. Hereby, the O atoms get plucked off and lost along the reaction, which is also affecting the dynamics. At the editor's discretion, and if the manuscript is not already considered too long, we would be happy to add the "predicted" $\delta15N$-N2O values with a short explanation.

―――――――――――――――――――

[Figure]

[Figure]

**Fig. 1.** N2O vs NO2- concentrations in (A) mineral plus dead biomass and (B) mineral only experiment

**A) N$_2$O Site Preference**

**B) $\delta^{15}$N$^{bulk}$-N$_2$O**

Mineral +
Dead Biomass

- ● 5.8
- ▼ 6.2
- ◆ 6.5
- ■ 6.9
- ▲ 7.1

**C) N$_2$O Site Preference**

**D) $\delta^{15}$N$^{bulk}$-N$_2$O**

Mineral only

- ● 5.8
- ▼ 6.2
- ◆ 6.5
- ■ 6.9
- ▲ 7.1

**Fig. 2.** Site Preference (SP; A, C) and $\delta$15Nbulk (B, D) values of N2O produced in experiments amended with mineral + dead biomass (red) and mineral-only (grey)

---

## Author Comment (AC2) · 3 Jul 2020

First, we would like to thank the reviewer for his/her valuable inputs and comments on our manuscript. We have to admit that the outliers in the N2O data are indeed real outliers due to a "concentration/linearity effect" during the measurement in which overly large peak areas in the raw data biased the results. After a thorough check of the raw data, these few data points were removed and the graphs were re-drawn. We contend the data now presented are valid and accurate. We apologize for the mistake.

[Figure]

L98: "hold the potential to disentangle abiotic and biotic NO2- reduction " - this cannot be concluded from the previous sentences, which say that for both biotic and abiotic processes we deal with significant isotope effect

Reply: We will rephrase that part.

L184: "flushed before for 5 hrs with 5.0 He" - is this right - you need to flush 5hrs? Why so long? Have you tested that this is needed?

Reply: Since we simply applied the flushing routine of the denitrifier method, the headspace vials were indeed flushed for 5 hrs. Later testing showed, that 3 hrs would also suffice. However, several hours of flushing seem to be necessary to reduce the blank value to acceptable levels, in particular when sample size is low.

L315: you mean Fig. 6 here?

Reply: We thank the reviewer for pointing this out and apologize for the mistake! Indeed, in L315 it should read Fig. 6. We will change this in the manuscript!

L315: Such a value seems rather not plausible, pleas double check your measurements and check how reliable is this value. There is no known process which could result in such negative value. Similarly, in 6C - I'd even doubt the value of -40 permil, unless you have ideas to explain this.

Reply: As already mentioned, we carefully checked the raw data as well as the corrected data files again and we have to admit that these values are indeed outliers caused by very high peak areas (concentration effect). We corrected the graphs accordingly (see Figure 1 "Site Preference (SP; A, C) and $\delta$15Nbulk (B, D) values of N2O produced in experiments amended with mineral + dead biomass (red) and mineral-only (grey)").

L346: Is further N2O reduction to N2 also possible? If not, please explain why.

Reply: Considering previous publications (Rivallan et al., 2009; Doane, 2017; Phillips

et al, 2016), an abiotic reduction of N2O to N2 is indeed possible, particularly in the presence of a reactive surface.

See L559-570: "Abiotic decomposition of N2O to N2 in the presence of Fe-bearing zeolites has been investigated previously (Rivallan et al., 2009). However, it remains unclear if this process could also occur here. Fractional N2O reduction is also not explicitly indicated by the SP values, which would reflect an increase with N2O reduction (Ostrom et al., 2007; Winther et al., 2018) [...] However, since N2O concentrations, even if minor, are increasing towards the end of the experiments, production and possible decomposition as well as ongoing sorption mechanisms might also serve as possible explanation leading to these rather low SP values."

However, with regards to the rather low N2O concentrations and given the relatively constant $\delta 15N_{bulk}$-N2O values, abiotic N2 production seems plausible. First, the N2O produced here accounts only for $\sim 0.7\%$ of the total NO2- reduced in the experiments. This large difference might be caused by sorption processes or simply by the fact that N2O is not the final product (Note: accumulation of the intermediates e.g. NO, is quite unlikely since they are extremely reactive). Furthermore, if N2O were indeed the final and only product, its $\delta 15N_{bulk}$ values should approximate the $\delta 15N$-NO2- values (starting off lighter than $\delta 15N$-NO2- and increasing over incubation time). However, here the $\delta 15N_{bulk}$-N2O values remained relatively steady or did not increase much throughout the experiment, which might indicate that N2O is not just produced but possibly also further reduced (multistep-reaction). Therefore, the production of N2, although abiotic, seems quite likely. We clarify this in the revised MS.

As written in L597-601: "Considering that the N2O concentrations measured in our experiments were comparatively low and that $\delta 15N_{bulk}$ N2O values did not noticeably change throughout the experiments, formation of N2 via abiotic interactions between NO2- and NO may also be involved (Doane, 2017; Phillips et al., 2016). Hence, N2O is possibly involved in the reaction either as an intermediate or as a side product, and can thereby influence the overall N and O isotope dynamics.".

L484: This is not clear: d15N decrease and initial decrease?

Reply: Here, we meant the decrease in $\delta$15N and an observed initial decrease in the concentration of NO2-. We will add "concentration" to avoid further confusion.

L547: "was calculated is based" - sentence to be rewritten

Reply: Again, we thank the reviewer for reading our manuscript so carefully. This will of course be corrected.

L548: What do the arrows mean? (in table 3)

Reply: The arrows were added to indicate an overall increase (arrow up) or decrease (arrow down) from the initial delta value. We will correct a mistake (line for $\delta$15N NO2- values - arrow for DB+mineral setup should point up) that we only now detected, and we will add the explanation in the caption of the table.

L614: This last sentence is not stated in the discussion - in discussion you just say it is unsure if abiotic N2 production is possible. Please explain this more detailed. It is not said in the discussion what is the isotope effect of abiotic N2O reduction to N2 (is this known?) - so I do not understand how N2O isotopic results can suggest its occurrence.

Reply: Generally, N2 production is still assumed to be caused mainly by enzymatic re- actions. However, there are studies providing evidence for abiotic N2 production (e.g., Rivallan et al., 2009; Phillips et al, 2016). In our manuscript, we choose to only cau- tiously refer to the possible abiotic N2O reduction to N2, since most N cycling studies still do not account for abiotic N2 production. Furthermore, our SP values do not explic- itly indicate the occurrence of fractional N2O reduction (N2O accumulates, SP values remain rather steady). Unfortunately, we did not analyse N2 samples, hence we do not know the range of N2 concentrations and/or isotope values, which would help to better address this aspect. To the best of our knowledge, the isotope effect of abiotic N2O reduction to N2 is unknown. As already mentioned above, N2O accumulates throughout the experiments but overall accounts only for a small fraction of the NO2-

reduced. Furthermore, the $\delta$15Nbulk-N2O values remained rather steady throughout the experiments, which indicates that other processes may influence the reaction dynamics and that N2O may simply be an intermediate. If, again, N2O were the final and only product, $\delta$15N-bulk values would be expected to increase with decreasing NO2- concentrations (and thus increasing $\delta$15N-NO2-). However, $\delta$15Nbulk-N2O values to not really change much toward the end of the experiments, and remain steady for quite some time. Thus they do not reflect the patterns expected for a final product.

[Figure]

A) N$_2$O Site Preference

B) $\delta^{15}N^{bulk}$-N$_2$O

Mineral +
Dead Biomass

— ● — 5.8
— ▼ — 6.2
— ◆ — 6.5
— ■ — 6.9
— ▲ — 7.1

C) N$_2$O Site Preference

D) $\delta^{15}N^{bulk}$-N$_2$O

Mineral only

— ● — 5.8
— ▼ — 6.2
— ◆ — 6.5
— ■ — 6.9
— ▲ — 7.1

**Fig. 1.** N2O vs NO2- concentrations in (A) mineral plus dead biomass and (B) mineral only
experiment

---

## Author Response (AR1)

**Author's response on the revised manuscript "Impact of reactive surfaces on the abiotic reaction between nitrite and ferrous iron and associated nitrogen and oxygen isotope dynamics" by Anna-Neva Visser et al.**

**1. Point-by-Point response to the reviews**

**1.1. Response to comments by Anonymous Referee #1**

First, we wish to thank the reviewer for his/her valuable inputs and comments on our manuscript.

L39-40: I'm surprised there are no older references to the role of iron.

Reply: We agree that indeed there are many more references regarding the role of iron in the environment. However, our choice can be considered as "best of" selection, covering a whole suite of different aspects: we choose (1) Expert et al., 2012 since they explicitly focus on the vital role of iron for all living organisms, its wide range of redox potentials and its catalytic role in various metabolic pathways; (2) Lovley et al., 1997, who reported on the importance of iron already in 1988, however, the publication chosen represents a nice "summary", focusing also on various reactions and thus its "remediative" capabilities. Obviously, we wanted to limit the number of references, but if the reviewer thinks of a specific publication, we will be happy to include it. Again, in light of the many publications on the importance of iron available, and since our manuscript is already very long, we simply decided to pick two references that support the statement/sentence.

General experiment setup section: The conditions of the experiment are anoxia and the addition of iron and nitrogen in the form of nitrite. Under these conditions, in the environment, it is conceivable that dissimilative reduction of nitrite to ammonium may occur. Of course under perfect abiotic conditions DNRA should not occur. Did the authors measure ammonium concentrations throughout the experiment to ensure that no other processes than the one under study were taking place?

Reply: As the reviewer stated, DNRA should not occur under abiotic conditions. Considering that the abiotic experiments were all performed under laboratory conditions, using a medium that contains already high amounts of ammonium (5.61 mM $NH_4Cl$, see 2.1), ammonium concentrations were only checked sporadically for some setups. Since only (if at all) minor fluctuations were observed, no further efforts to determine ammonium concentrations were attempted.

L120-121: How long does it take from incubation to the measurement of concentrations and isotopes? Light is a factor that can generate abiotic reactions, which in turn can generate isotope fractionation. What about it?

Reply: Yes, light-induced reactions have to be considered. That was one reason why nitrite concentrations were measured via CFA immediately after the samples were taken (within one hour). After determining the nitrite concentrations, the azide method was applied (within max. 2-3 hrs). Samples were kept inside the glovebox in coloured (dark brown or blue) Eppendorf tubes, whereas the latter were chosen to inhibit potential photocatalytic reactions. The azide-treated headspace vials were stored in card boxes at RT until measured. At this point, the sample is fixed (i.e., turned into $N_2O$). Therefore, we are rather confident that neither light nor (possibly) temperature could have influenced the values. However, one could argue that the blue coloured Eppendorf tubes might not suffice, since they are indeed partly translucent. Since during one of the experiments blue and brown vials were used, and still, the concentration values within the nine replicates were very similar (see Figure 1 A and C, note error bars), we are confident that the rapid processing and precautions taken to avoid light-induced reactions did indeed suffice.

L179-180: Two nitrite isotope standards have been used. What are the values of these standards? Do these values include those of the samples measured in this study? What is the analytical precision of the method (preparation + intrinsic analysis) for the determination of the isotopic composition of nitrite (15N and 18O)?

Reply: Standard N-7373 has a $d^{15}N$ value of -79.6‰ and a $\delta^{18}O$ value of +4.5‰. In contrast, standard N-10219 has a $\delta^{15}N$ value of +2.8‰ and a $\delta^{18}O$ value of +88.5‰. Using both standards allowed for the reliable correction using standard bracketing: The standard $\delta^{15}N$ range included the $\delta^{15}N$ values obtained for our samples perfectly. The $\delta^{18}O$ values measured fell only slightly below (-0.5 to 2.5‰) the range given by the standards, so that corrections are reliable. Based on replicate measurements of laboratory standards and samples, the analytical precision for $NO_2^-$ $\delta^{15}N$ and $\delta^{18}O$ analyses was ±0.4‰ and ±0.6‰ (1 SD), respectively.

L285-291: Rayleigh conditions allow the isotope fractionation factor to be easily determined by looking at the slope of the line on a representation ln C/C0 as a function of d15N, but not C (with C the concentration at time t and C0 the initial concentration). This paragraph is not clear to me. Moreover, doesn't the fact that there is first a decrease of 15N, i.e. an inverse isotopic fractionation, with a decrease of the amount of heavy isotope in the residual substrate, and then an enrichment, mean that several processes could take place and that process 1 takes place at the beginning of the experiment with a higher rate than the second process which either starts at the beginning of the experiment or when process 1 is completed? Very concretely, the trend line is calculated on the points starting from the lowest d15N values? I think it would be necessary to clarify this part.

Reply: We agree, the title of the x-axis of Figure 5 might be misleading. Of course, the values of the x-axis represent the ln of the substrate fraction remaining (as mentioned in the caption below the figure). Hence, it is the ln(f) whereas f is C/C0. We will change the title of the x-axis to avoid future confusions. With regards to the second comment, i.e., that the data presented might simply reflect that two different processes are at work, we also agree. However, since it is hard to explain which processes might be at work and if this is indeed a clear inverse effect, we decided to calculate the isotope effect using the lowest $\delta^{15}N$ values observed (i.e. for the experimental period where we show a clear decline in nitrite concentration with a net increase in $\delta^{15}N$). We will clarify that there is putative evidence for multiple processes occurring in the incubations, and that this has implications for the Rayleigh approach.

L296-302: Is it not possible to envisage that the variations in 18O are due solely to an exchange between the oxygen of the nitrite and the oxygen of the water? By the way, what is the isotopic composition of water? Is it constant during the experiment?

Reply: Unfortunately, the isotopic composition of the water was not measured, and we can only assume its $\delta^{18}O$ (the water used in Tübingen has a $\delta^{18}O$ of roughly 11‰). It is possible that the variations in $\delta^{18}O$ are partially attributable to oxygen atom exchange dynamics with the matrix (see e.g. L504-516). However, considering that the observed drop in $\delta^{18}O$ values in both experiments occurs more or less simultaneously with the drop in $\delta^{15}N$ might be indicative of other dynamics (e.g. sorption, complexation?).

L309-313: The authors have done a significant analytical work. Why not show the variations in N2O concentration as a function of nitrite concentrations? Before any interpretation with isotopes or isotopologists, it seems to me useful and necessary to work on the concentrations and in particular to make mass balances.

Reply: The proposed graph could be added to the supplementary material. However, particularly for the mineral only setups, this way of visualizing the data does not help much (see graph added). Also, for the main manuscript we had severe concerns with regards to its length. Therefore, we chose to present only graphs that really help to understand the main messages of this project. With regards to the mass balance: The initial objectives of this project included mass balance considerations since it was supposed to lay the ground for a following study on nitrate-dependent Fe(II) oxidation in selected microbial strains. Unfortunately, we did not have the capacities to also analyse the $N_2$ samples, so a proper mass balance is unfortunately not possible.

[Figure]

**Figure 1: N₂O vs NO₂⁻ concentrations in (A) mineral plus dead biomass and (B) mineral only experiment**

L314-315: The authors do not discuss the very negative SP value, which is very distinct from the other points. Is this an analytical problem?

Reply: We assume that the reviewer is referring to the observed drops in SP values (-120 to -80‰), occurring at $t_1$ for samples taken from the mineral + dead biomass setup at pH 6.2 and mineral only at pH 5.8. After another thorough check of the raw data, we have to admit that for those particular samples the peak areas of the data obtained via CF-IRMS were much higher (compared to standards), possibly causing an extreme linearity or contamination effect that is affecting the data. We re-checked the entire data set again and removed these outliers (see revised figure below). The bulk of the data is not compromised, as we have good agreement between the standard and the sample peak areas.

[Figure]

**Figure 2: Site Preference (SP; A, C) and δ¹⁵N^bulk (B, D) values of N₂O produced in experiments amended with mineral + dead biomass (red) and mineral-only (grey)**

L326: There is no figure S6. But mentioned in S5 section figure 3.

Reply: We thank the reviewer for pointing this out and apologize for the mistake. Figure S5 mentioned in L322 actually corresponds to Figure S4 in the supplements, while S6 in L 326 refers to S5! We will change this in the re-submitted version of the MS.

L484-486: Large variations of δ15N are not associated with variations of δ18O. While these are measurements made on the residual substrate. The drop in ¹⁸O at the beginning of the experiment is more likely due to an isotopic exchange with the oxygen in the water than evidence of a process.

Reply: Whether the drop is solely caused by the O isotopic exchange or, maybe partially, by interactions with the mineral surface, is not really clear. The drop observed in δ¹⁸O occurs almost simultaneously with the e.g. the decrease in δ¹⁵N for the mineral + dead biomass experiment. This might be indicative of other processes playing indeed a certain role. However, as we tried to explain in L496ff in the original MS, we assume that the main effect is the oxygen exchange with the water of the medium, which simply takes time and thus results in "fluctuations" (especially for the mineral only experiments) until the entire system is equilibrated.

L531-538: It might be interesting to look at δ18O variations of N2O during the experiment. And see if it correlates with that of nitrite. This would also be an opportunity to confirm or deny whether there is an isotope exchange between the oxygen in the nitrite and the oxygen in the water.

Reply: Indeed, using the δ¹⁸O variations of N₂O might help to better understand the isotope exchange processes within the system. However, since N₂O is definitely not the only product and possibly further reduced (resulting in a branching effect caused by the removed O atoms, which is further affecting the O dynamics within the system), this approach would be biased.

L551-552: if N2O is considered to accumulate, it can be considered to be the accumulated product in the case of a Rayleigh distillation. In this case, and taking into account the isotope fractionation associated with nitrite reduction, it is easy to calculate what the expected 15N and 18O of the N2O produced. It would then be interesting to compare the measured values with the expected values.

Reply: We agree that it is indeed possible to estimate the predicted value of $\delta^{15}N$ by using the accumulated product equation. An epsilon value calculated from the $\delta^{15}N\text{-}NO_2^-$ data could be used to estimate the predicted $\delta^{15}N\text{-}N_2O$ values, which would be different since $N_2O$ is clearly not the single product. However, for $\delta^{18}O$ this approach would not work due to the branching effect occurring during reduction. Hereby, the O atoms get plucked off and lost along the reaction, which is also affecting the dynamics.

At the editor's discretion, and if the manuscript is not already considered too long, we would be happy to add the "predicted" $\delta^{15}N\text{-}N_2O$ values with a short explanation.

**1.2. Response to comments by Anonymous Referee #2**

First, we would like to thank the reviewer for his/her valuable inputs and comments on our manuscript. We have to admit that the outliers in the $N_2O$ data are indeed real outliers due to a "concentration/linearity effect" during the measurement in which overly large peak areas in the raw data biased the results. After a thorough check of the raw data, these few data points were removed and the graphs were re-drawn. We contend the data now presented are valid and accurate. We apologize for the mistake.

L98: "hold the potential to disentangle abiotic and biotic NO2- reduction " - this cannot be concluded from the previous sentences, which say that for both biotic and abiotic processes we deal with significant isotope effect

Reply: We will rephrase that part.

L184: "flushed before for 5 hrs with 5.0 He" - is this right - you need to flush 5hrs? Why so long? Have you tested that this is needed?

Reply: Since we simply applied the flushing routine of the denitrifier method, the headspace vials were indeed flushed for 5 hrs. Later testing showed, that 3 hrs would also suffice. However, several hours of flushing seem to be necessary to reduce the blank value to acceptable levels, in particular when sample size is low.

L315: you mean Fig. 6 here?

Reply: We thank the reviewer for pointing this out and apologize for the mistake! Indeed, in L315 it should indeed read Fig. 6. We will change this in the manuscript!

L315: Such a value seems rather not plausible, pleas double check your measurements and check how reliable is this value. There is no known process which could result in such negative value. Similarly, in 6C - I'd even doubt the value of -40 permil, unless you have ideas to explain this.

Reply: As already mentioned, we carefully checked the raw data as well as the corrected data files again and we have to admit that these values are indeed outliers caused by very high peak areas (concentration effect). We corrected the graphs accordingly (see graph attached).

[Figure]

**Figure 3:** *Same as Figure 2* - Site Preference (SP; A, C) and $\delta^{15}N^{bulk}$ (B, D) values of $N_2O$ produced in experiments amended with mineral + dead biomass (red) and mineral-only (grey)

L346: Is further N2O reduction to N2 also possible? If not, please explain why.

Reply: Considering previous publications (Rivallan et al., 2009; Doane, 2017; Phillips et al, 2016), an abiotic reduction of $N_2O$ to $N_2$ is indeed possible, particularly in the presence of a reactive surface.

See L559-570: "Abiotic decomposition of $N_2O$ to $N_2$ in the presence of Fe-bearing zeolites has been investigated previously (Rivallan et al., 2009). However, it remains unclear if this process could also occur here. Fractional $N_2O$ reduction is also not explicitly indicated by the SP values, which would reflect an increase with $N_2O$ reduction (Ostrom et al., 2007; Winther et al., 2018) [...] However, since $N_2O$ concentrations, even if minor, are increasing towards the end of the experiments, production and possible decomposition as well as ongoing sorption mechanisms might also serve as possible explanation leading to these rather low SP values."

However, with regards to the rather low $N_2O$ concentrations and given the relatively constant $\delta^{15}N^{bulk}$-$N_2O$ values, abiotic $N_2$ production seems plausible. First, the $N_2O$ produced here accounts only for ~0.7% of the total $NO_2^-$ reduced in the experiments. This large difference might be caused by sorption processes or simply by the fact that $N_2O$ is not the final product (Note: accumulation of the intermediates e.g. NO, is quite unlikely since they are extremely reactive). Furthermore, if $N_2O$ were indeed the final and only product, its δ15Nbulk values should approximate the $\delta^{15}N$-$NO_2^-$ values (starting off lighter than $\delta^{15}N$-$NO_2^-$ and increasing over incubation time). However, here the $\delta^{15}N^{bulk}$-$N_2O$ values remained relatively steady or did not increase much throughout the experiment, which might indicate that $N_2O$ is not just produced but possibly also further reduced (multistep-reaction). Therefore, the production of $N_2$, although abiotic, seems quite likely. We clarify this in the revised MS.

As written in L597-601: "Considering that the $N_2O$ concentrations measured in our experiments were comparatively low and that $\delta^{15}N^{bulk}$-$N_2O$ values did not noticeably change throughout the experiments, formation of $N_2$ via abiotic interactions between $NO_2$- and $NO$ may also be involved (Doane, 2017; Phillips et al., 2016). Hence, $N_2O$ is possibly involved in the reaction either as an intermediate or as a side product, and can thereby influence the overall N and O isotope dynamics.".

L484: This is not clear: d15N decrease and initial decrease?

Reply: Here, we meant the decrease in $\delta^{15}N$ and an observed initial decrease in the concentration of $NO_2^-$. We will add "concentration" to avoid further confusion.

L547: "was calculated is based" - sentence to be rewritten

Reply: Again, we thank the reviewer for reading our manuscript so carefully. This will of course be corrected.

L548: What do the arrows mean? (in table 3)

Reply: The arrows were added to indicate an overall increase (↑) or decrease (↓) from the initial delta value. We will correct a mistake (line for $\delta^{15}N$-$NO_2^-$ values - arrow for DB + mineral setup should point up) that we only now detected, and we will add the explanation in the caption of the table.

L614: This last sentence is not stated in the discussion - in discussion you just say it is unsure if abiotic N2 production is possible. Please explain this more detailed.

It is not said in the discussion what is the isotope effect of abiotic N2O reduction to N2 (is this known?) - so I do not understand how N2O isotopic results can suggest its occurrence.

Reply: Generally, $N_2$ production is still assumed to be caused mainly by enzymatic reactions. However, there are studies providing evidence for abiotic $N_2$ production (e.g. Rivallan et al., 2009; Phillips et al, 2016). In our manuscript, we choose to only cautiously refer to the possible abiotic $N_2O$ reduction to $N_2$, since most N cycling studies still do not account for abiotic $N_2$ production. Furthermore, our SP values do not explicitly indicate the occurrence of fractional $N_2O$ reduction ($N_2O$ accumulates, SP values remain rather steady). Unfortunately, we did not analyse $N_2$ samples, hence we do not know the range of $N_2$ concentrations and/or isotope values, which would help to better address this aspect.

To the best of our knowledge, the isotope effect of abiotic $N_2O$ reduction to $N_2$ is unknown. As already mentioned above, $N_2O$ accumulates throughout the experiments but overall accounts only for a small fraction of the $NO_2^-$ reduced. Furthermore, the $\delta^{15}N^{bulk}$-$N_2O$ values remained rather steady throughout the experiments, which indicates that other processes may influence the reaction dynamics and that $N_2O$ may simply be an intermediate. If, again, $N_2O$ were the final and only product, $\delta^{15}N^{bulk}$ values would be expected to increase with decreasing $NO_2^-$ concentrations (and thus increasing $\delta^{15}N$-$NO_2^-$). However, $\delta^{15}N^{bulk}$-$N_2O$ values to not really change much toward the end of the experiments, and remain steady for quite some time. Thus they do not reflect the patterns expected for a final product.

**2. List of relevant changes**

**2.1. Adjustments according to our responses to comments by anonymous Referee #1**

| | |
|---|---|
| L147 | Sampling procedure details added; "within one hour after the sample was taken via a…" |
| L150 | Ferrozine analysis details added; "SFA- and/or HCl-fixed samples were stored in the dark and at 4°C until" |
| L153 | Procedure details added; "Triplicate samples" |

| L179f | Procedure details added; "…upside down at room temperature and in the dark. Two nitrite isotope standards, namely (N-7373 ($\delta^{15}$N: -79.6‰, $\delta^{18}$O: +4.5‰) and N-10219 ($\delta^{15}$N: +2.8‰; $\delta^{18}$O; +88.5‰); (Casciotti & McIlvin, 2007)…" |
|---|---|
| L182 | Sentence added: "Based on replicate measurements of laboratory standards and samples, the analytical precision for $NO_2^-$ $\delta^{15}$N and $\delta^{18}$O analyses was ±0.4‰ and ±0.6‰ (1 SD), respectively." |
| L213 | Added reference to Figure S4 added (S4 – requested Figure, added to the supplementary information) |
| L303 | Figure 5, x-axis title changed to "ln (f)" |
| L316 | Figure 6 replaced with a corrected version; Caption changed to "…For pH 6.5, the final SP value (A) is missing due to analytical problems (overly large sample peak areas). Standard error calculated from biological replicates (n = 3 or 2 ) is represented by the error bars." |
| L321-327 | References to Figures S5 and S6 changed to S6 and S7, respectively |
| L598ff | Changed to "Considering that the $N_2O$ concentrations measured in our experiments were comparatively low and that $\delta^{15}N^{bulk}$-$N_2O$ values did not noticeably change throughout the experiments, it is **unlikely that $N_2O$ is the final product**, and formation of $N_2$ via abiotic interactions between $NO_2^-$ and NO is probably also involved (Doane, 2017; Phillips et al., 2016). Indeed, if **accumulated as the final product, the $\delta^{15}N^{bulk}$- $N_2O$ value at the end of the incubation should be ~-33‰ (according to closed-system accumulated-product Rayleigh dynamics), significantly higher than what we measured (~ -50 ±6 ‰)**. Hence, whether $N_2O$ is an intermediate or parallel side product, its role in the overall reaction complicates N and O isotope mass balance dynamics in complex ways." |

**2.2. Adjustments according to our responses to comments by anonymous Referee #2**

| L98-100 | "This suggests that coupled N and O isotope measurements hold the potential to disentangle abiotic and biotic NO2- reduction in the presence of Fe(II)." changed to "However, reaction kinetics can significantly affect isotope reaction dynamics, and chemodenitrification is possibly impacted by e.g. concentration effects and/or the presence of different catalysts (i.e. surfaces, organics). Hence, performing coupled N and O isotope measurements might help to gain deeper insights into the mechanistic details and fractionation systematics of $NO_2^-$ reduction in the presence of Fe(II)." |
|---|---|
| L315 | "(Figure 5 A,C)" replaced by "(Figure 6 A, C)" |
| L316 | Figure 6 replaced with a corrected version; Caption changed to "…For pH 6.5, the final SP value (A) is missing due to analytical problems (overly large sample peak areas). Standard error calculated from biological replicates (n = 3 or 2 ) is represented by the error bars." |
| L484f | "…was observed with the initial decrease…" changed to "…occurred in parallel contemporaneously with initially decreasing in $NO_2^-$ concentrations." |
| L545ff | Table 3 – Caption corrected (plus values): "$\delta^{15}$N and $\delta^{18}$O values were calculated using $\overline{x}_{t0} - \overline{x}_{tend}$. Isotope fractionation was calculated is based on the slope between the lowest initial value (here at t1) and tend for all pH." changed to "$\delta^{15}$N and $\delta^{18}$O values were calculated using $\overline{x}_{t0} - \overline{x}_{tend}$, whereas an overall increase from the initial value is marked with ↑, and a decrease with ↓. The calculated isotope fractionation factor ($\varepsilon$) is based on the slope between the lowest initial value (here at $t_1$) and $t_{end}$ for all pH." |
| L598ff | Changed to "Considering that the $N_2O$ concentrations measured in our experiments were comparatively low and that $\delta^{15}N^{bulk}$-$N_2O$ values did not noticeably change throughout the experiments, **it is unlikely that $N_2O$ is the final product, and formation of $N_2$ via abiotic interactions between $NO_2^-$ and NO is probably also involved** (Doane, 2017; Phillips et al., 2016). Indeed, if accumulated as the final |

product, the $\delta^{15}N^{bulk}$-$N_2O$ value at the end of the incubation should be ~-33‰ (according to closed-system accumulated-product Rayleigh dynamics), significantly higher than what we measured (~ -50 ±6 ‰). Hence, whether $N_2O$ is an intermediate or parallel side product, its role in the overall reaction complicates N and O isotope mass balance dynamics in complex ways."

**2.3. General Adjustments**

| | |
|---|---|
| L48 | L55 to 65 moved upwards, removed L48 to L50 |
| L50f | Sentence merged with first part of the next sentence |
| L53 | Added "EPS has been demonstrated to…" |
| L55 | "biologically" changed to "enzymatically" |
| L70 | Reference added (Zhu-Barker et al., 2012) |
| L203 | Figure 1: Caption corrected – pH 5.**8** |
| L235 | "lost" changed to "processing failed" |
| L251 | Added "sample processing failed for the", removed "was lost" |
| L291 | Added reference to Figure 4 C |
| L295 | (Figure S4) Rayleigh plot for mineral only experiments now added to Supplementary information file |
| L309 | "amended" replaced with "mineral plus DB"; "(SP)" added after "Site preference" |
| L314f | SP values in text replaced with corrected values |
| L356-358 | Sentence deleted |
| L451-454 | Sentence deleted |
| L532 | "…(abiotic -46.5 ±0.2‰; dead biomass -49.4 ±1.0‰)…" changed to "…(abiotic -49.5 ±0.6‰; dead biomass -50.5 ±0.8‰)…" |
| L555 | "mineral-only treatment (27.9‰) is only slightly higher than that of the DB experiment (23.2‰)," changed to "mineral-only treatment (30.9‰) is slightly higher than that of the DB experiment (24.4‰)" |
| L562f | "relatively low (6.0 ± 0.8‰; 1.7 ± 1.2‰; Fig. 6) " changed to "relatively low (6.5 ± 0.8‰; 2.3 ± 1.2‰; Fig. 6, Table 3)." |
| L602 | Figure 8 slightly corrected (colours of bonds between species) |
| L661-664 | Acknowledgements corrected (added: Toby Samuels and Louis Rees) |
| L675ff | Changed formatting of the reference list |
| Supplements | S4 to S7 were corrected (L20: now S4 – graph depicting $N_2O$ versus $NO_2^-$ concentrations, requested by referee#1; L24 now S5 – Rayleigh plots for the mineral-only setups; L29: now S6 – Rayleigh plots for $N_2O$ $\delta^{15}N^{\alpha}$, $\delta^{15}N^{bulk}$ and site preference, SP; L34: now S7 – Plot showing $\delta^{18}O$ vs $\delta^{15}N^{bulk}$ in $N_2O$ for mineral-only and mineral plus dead biomass setups) |

[revised manuscript text omitted]

**Kommentiert [AV6]:** Used:
$D15N_{PA} = d15N_{S,t0} - 15e*f*\ln(f)/(1-f)$
See
https://www.whoi.edu/cms/files/jhayes/2005/9/IsoCalcs30Sept04_51
83.pdf
Equation 46
Or Casciotti 2011 Equation 11.3

dynamics), which  is significantly  higher than what we have measured (∼ -50 ±6 ‰).
Hence,
seems that N₂O is likely to  meddl with the overall reaction dynamics either as an
intermediate or as a side product, and can thereby influence the overall N and O isotope dynamics in highly complex ways.

[Figure]

**Figure 8: Conceptual figure depicting the proposed reaction mechanisms and feedbacks between the different N species during**
**chemodenitrification induced by the presence of a mineral surface (lower left corner) or (dead) biomass (upper right corner).**
**Adsorption of Fe²⁺ (directly or via complexation by OH⁻) as well as NO₂⁻ could catalyse a direct reaction between both. In addition,**
**NO₂⁻ adsorption onto the Fe(II) mineral might also induce disproportionation, leading to NOₓ formation. These formed**
**intermediates, although transitory, may impact the overall reaction dynamics by e.g. complex formation (i.e. [NO--Fe²⁺]) or direct**
**Fe(II) oxidation. The produced Fe(III) might induce another feedback loop (autocatalysis) resulting in further Fe(II) oxidation.**
**Similar processes are possibly induced by the presence of (dead) biomass. Adsorption and complexation of either NO₂⁻ and Fe²⁺**
**would enhance the reaction between both. In addition, the presence of organic acids would decrease the pH locally and thereby**
**promote and accelerate NO₂⁻ disproportionation and thus additionally enhance Fe(II) oxidation. Our results suggest that NO₂⁻**
**reduction results in an KIE, which should influence the isotopic composition of NO. N₂O here is an intermediate, the isotopic**
**composition of which is mainly influenced by an EIE between NO and N₂O. The low N₂O yields as well as the N₂O isotopic results**
**(bulk, SP) clearly suggests that N₂ is produced abiotically.**

**Kommentiert [ML7]:** This does not relly help and improve things…it is not clear what you want to say here.

**Kommentiert [ML8]:** We do not need this to make the point

**Kommentiert [AV9]:** Corrected (colours were missing for some of the bonds)

[revised manuscript text omitted]

**Kommentiert [ML10]:** Check refs…they seem unformatted, some are messed up of only in capital letters (e.g. Widdel)